# Trajectory-LLM: A Language-based Data Generator for Trajectory Prediction in Autonomous Driving

**Kairui Yang**[1][*]  **Zihao Guo**[1][*]  **Gengjie Lin**[2]  **Haotian Dong**[1][†]  **Zhao Huang**[1]
**Yipeng Wu**[1]  **Die Zuo**[1]  **Jibin Peng**[1]  **Ziyuan Zhong**[3]  **Xin Wang**[4]  **Qing Guo**[5]
**Xiaosong Jia**[2]  **Junchi Yan**[2]  **Di Lin**[1][†]

[1]Tianjin University   [2]Shanghai Jiaotong University   [3]Columbia University
[4]The Hong Kong Polytechnic University
[5]IHPC and CFAR, Agency for Science, Technology and Research, Singapore

## ABSTRACT

Vehicle trajectory prediction is a crucial aspect of autonomous driving, which requires extensive trajectory data to train prediction models to understand the complex, varied, and unpredictable patterns of vehicular interactions. However, acquiring real-world data is expensive, so we advocate using Large Language Models (LLMs) to generate abundant and realistic trajectories of interacting vehicles efficiently. These models rely on textual descriptions of vehicle-to-vehicle interactions on a map to produce the trajectories. We introduce Trajectory-LLM (Traj-LLM), a new approach that takes brief descriptions of vehicular interactions as input and generates corresponding trajectories. Unlike language-based approaches that translate text directly to trajectories, Traj-LLM uses reasonable driving behaviors to align the vehicle trajectories with the text. This results in an "interaction-behavior-trajectory" translation process. We have also created a new dataset, Language-to-Trajectory (L2T), which includes 240K textual descriptions of vehicle interactions and behaviors, each paired with corresponding map topologies and vehicle trajectory segments. By leveraging the L2T dataset, Traj-LLM can adapt interactive trajectories to diverse map topologies. Furthermore, Traj-LLM generates additional data that enhances downstream prediction models, leading to consistent performance improvements across public benchmarks. The source code is released at `https://github.com/TJU-IDVLab/Traj-LLM`.

## 1 INTRODUCTION

Accurately predicting the trajectory of vehicles is a crucial aspect of autonomous driving systems. A vast amount of data regarding vehicle trajectories is required to train the trajectory prediction models. However, collecting such data from real-world scenarios requires considerable manual effort and resources. This results in significant costs associated with capturing vehicle trajectories that exhibit intense interactions like overtaking, yielding, and bypassing.

The autonomous driving industry has developed a solution of using traffic-flow generators to synthesize trajectories and enhance training and testing data. An excellent generator should generate much synthetic data with minimal human operation. Previous works (Li et al., 2023; Feng et al., 2024) have shown the effectiveness of combining real-world and synthetic data for training prediction models. Popular generators like LGSVL (Rong et al., 2020) have ***graphical interfaces*** that enable users to create and control vehicle trajectories and interaction relationships on a map through drag-and-drop operations. However, even a single vehicle's precise editing often requires tens of drag-and-drop operations. Other generators like Scenic (Fremont et al., 2023) provide ***programmable interfaces*** that allow users to control vehicle motion using functional parameters such as position, velocity, and yaw. However, they require users to have excellent programming skills to utilize codes to depict complex vehicle interactions.

The recent literature advocates a more user-friendly generator with ***language interfaces*** built on top of the large language models (LLMs) (Zhong et al., 2023a;b; Tan et al., 2023; Ding et al., 2024;

---

[*]Co-first authors.
[†]Co-corresponding authors.

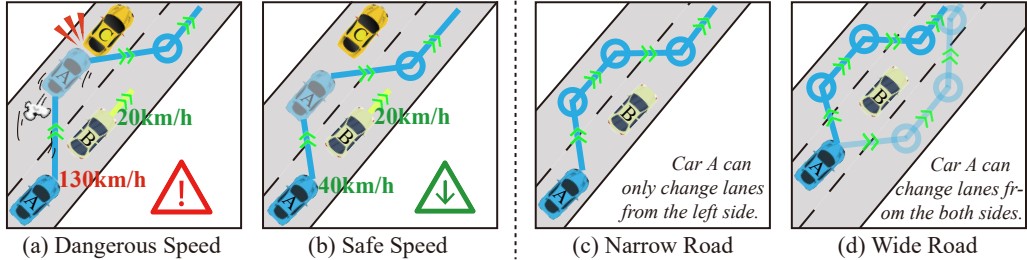

Figure 1: (a) Dangerous speeding of Car A for overtaking a slow Car B is unusual, which may lead to a collision with Car C. (b) A reasonable speed of Car A for overtaking a slow Car B follows a safer driving logic. Without the guidance of driving logic, a language interface that has only seen trajectories of left turns in narrow road scenes like (c) may find it difficult to generate trajectories such as left and right turns in wide road scenes like (d).

Mao et al., 2023; Zhao et al., 2024), which translates the brief text of vehicle interactions to the trajectories in the map (a.k.a., "interaction-trajectory" translation). The brevity of the text precludes the specific parameters regarding vehicle motions, rendering it a highly abstract description of vehicle interactions. Therefore, users can efficiently generate many vehicle trajectories that align with the interaction described by the text. **This efficiency motivates us to study the vehicle trajectory generator based on the large language model for saving data collection efforts.** The vehicle trajectories can be viewed as a collection of waypoints, with each waypoint associated with the specific motion parameters of the vehicle as it passes through. These trajectories concretely describe the vehicle interaction. Language interfaces utilize the map's environmental information and vehicles' partial trajectories as guidance to generate the complete trajectories aligned with the text. This approach is similar to trajectory prediction models, but these two tasks fundamentally differ. While trajectory prediction models aim to predict relatively deterministic vehicle paths, using language interfaces may generate many trajectories that align with the interaction described in the short text.

However, when delving deeper into the vehicle trajectories generated by the language interface, many of them show unreasonable driving behaviors (see Figure 1). In contrast to the brief text of vehicle interaction and the concrete parameters of vehicle motion, the driving behaviors (e.g., change lane, cruise, accelerate/decelerate, and stop) reflect the inherent logic of human beings or autonomous driving systems that dynamically adjust the vehicle motions based on the changing environment. In Figure 1(a–b), given the text description of "Car A overtakes a very slow Car B", the trajectory of Car A in (a) represents a dangerous behavior of speeding, while in (b) Car A overtakes B at a safe speed. This leads to a lack of realism in the generated data. Furthermore, without reasonable driving behaviors, the language interface may become accustomed to generating trajectories like those in the known scenarios rather than the novel ones in the unseen scenarios, thus lowering the data diversity. In Figure 1(c–d), given the text of "Car A bypasses a stationary Car B", Car A in (c) can only change lanes from the left side to bypass B on a narrow road after extensive training, but failing to generate a legal right-hand bypassing on an unseen wide road in (d). **Thus, the primary objective of this paper is to involve the human driving logic to guide the translation between the text description of vehicle interactions and trajectories.**

This paper reformulates the "interaction-trajectory" translation by the language interface to the "interaction-behavior-trajectory" translation by adding the driving behaviors in-between to guide the generation of vehicle trajectories. We propose a Trajectory-LLM (Traj-LLM), a language-based generator for producing vehicle trajectories. As illustrated in Figure 2, Traj-LLM divides the "interaction-behavior-trajectory" translation into two stages. At the first stage (see Figure 2(a)), Traj-LLM takes input as the text of interaction between multiple vehicles and the map with lanes, outputting the driving behaviors of each vehicle. Chronologically, we organize the driving behaviors of each vehicle into a text sequence. Each driving behavior is followed by a text that describes the logic behind this behavior, in the format like "Change Lane: There are approximately 5 meters of road width on the left, while the road width on the right is insufficient, therefore choosing to change to the left lane" and "Change Lane: Both the left and right sides have ample road widths, allowing for a lane change to either the left or the right". At the second stage (see Figure 2(b)), Traj-LLM fuses the vehicle interactions and behaviors in the text with the map information to output specific motion parameters for each vehicle's trajectory.

We introduce a dataset, named ***Language-to-Trajectory*** (L2T), to aid in the training of Traj-LLM and associated research efforts. L2T comprises vehicle trajectories derived from 240K traffic scenarios, encompassing six diverse road topologies such as straightway, bend, roundabout, cross/T-shaped/Y-shaped intersection. Skilled drivers carefully craft these intricate trajectories and utilize textual segments to capture the vehicle interactions and behaviors within each scenario. We engage our drivers to provide annotations regarding vehicle interactions and behaviors for these scenarios.

Traj-LLM can generate realistic, diverse, and interactive vehicle trajectories based on brief texts. These generated trajectories can be utilized for training trajectory prediction models, significantly improving their performances. We summarize our contributions as follows:

- We advocate a new paradigm of the language interface with the "interaction-behavior-trajectory" translation to generate trajectories coherent with the human driving logic.

- We propose a novel language interface named Traj-LLM, a vehicle trajectory generator based on the large language model. Furthermore, we collect a new L2T dataset containing 240K traffic scenarios with vehicles' interactive trajectories. This dataset also contains rich text descriptions of vehicle behaviors and interactions for training Traj-LLM's "interaction-behavior-trajectory" translation.

- Traj-LLM trained on the L2T dataset can generate vehicle trajectories as additional data for training trajectory prediction models, whose performances are effectively improved on the public Waymo and Argoverse datasets. These results can further inspire the relevant research on the language-based trajectory generator.

## 2    RELATED WORK

We survey three groups of generators, which are equipped with the graphical, programmable, and language interfaces to control the vehicle interactions and generate the trajectories.

**Graphical Interface**   Generators with graphical interfaces can be divided into Logsim and Worldsim. Logsim uses the real scenes to provide complex traffic situations. The scenes provided by Logsim are unchangeable, thus it fails to test the object interactions specified by different users. Worldsim allows users to design the scenes for testing the autonomous vehicles. These generators (Shah et al., 2018; Samak et al., 2023; Dosovitskiy et al., 2017; Rong et al., 2020; Silvera et al., 2022) employ the game engines like Unity 3D [1] that have the graphic renders of the realistic scenes obeying the real-world rules. Nevertheless, Worldsim costs expensive labor to operate the graphical interface to control every vehicle to create interactions. In contrast, the language interface relies on the short text to efficiently create the object interactions.

**Programmable Interface**   Compared to the graphical interface that partially omits the details of vehicle motions to simplify the generator, the programmable interfaces (Zhao et al., 2024; Yang et al., 2023) offer an extensive collection of controlling functions of the vehicle interactions. Scenic (Fremont et al., 2023) is a domain-specific languages for describing the vehicle behaviors. Yet, they need a significant amount of code to build a road scene with complex vehicle interactions. The Reinforcement learning (Rempe et al., 2023; Janner et al., 2022; Li et al., 2019; Shah et al., 2018; Dosovitskiy et al., 2017; Zhang et al., 2021; Rong et al., 2020; Amini et al., 2020; Zhong et al., 2023b) allows the traffic rules, which also belong to the formal language, to guide the generation of the specific vehicle interactions without requiring intricate coding. The traffic rules help to construct the reward function that drives reinforcement learning. The limited number of traffic rules not only constrain the driving behavior of each vehicle, but fail to express the logic of making any behavior. This logic is important for teaching vehicles to navigate different maps, executing maneuvers such as lane changes and merges safely.

In this work, we introduce driving logic applicable to different map topologies through text. These logics can guide LLM to generate reasonable vehicle trajectories in various map topologies, thereby increasing the diversity of generated data. Diverse data aids in training downstream trajectory prediction models, enhancing prediction accuracy in complex scenarios.

---

[1]https://unity.com/

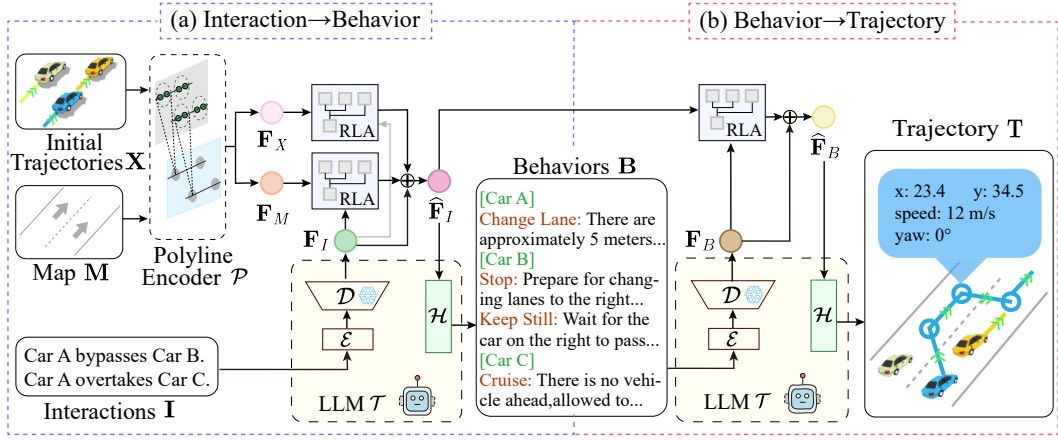

Figure 2: The two stages of the "interaction-behavior-trajectory" translation. (a) We employ LLM with the random locality attention to translate the textual description of vehicle interactions into the behavior of each vehicle. Each behavior is associated with the underlying logic. (b) Given the vehicle interactions and behaviors, LLM translates them to the sequential motion parameters that represent the trajectory of each vehicle. We illustrate the random locality attention in Figure 3.

**Language Interface**  The recent LLMs (Devlin et al., 2019; Stiennon et al., 2020; Brown et al., 2020; Touvron et al., 2023; Bai et al., 2023) enable the language interface to create virtual scenes with complicated vehicle interactions. Though people can use the text-to-video methods (Blattmann et al., 2023; Wu et al., 2023; Ho et al., 2022; Voleti et al., 2022) and traffic generation methods (Park et al., 2023; Zhong et al., 2023a;b; Achiam et al., 2023; Vemprala et al., 2024; Cui et al., 2024; Wen et al., 2023; Chen et al., 2024; Mao et al., 2023; Sha et al., 2023; Jin et al., 2023) to design the object interactions, they may fail to produce the usual vehicle motions like those in the real world. CTG (Zhong et al., 2023b) and CTG++ (Zhong et al., 2023a) depend on the traffic rules to generate realistic motions for the individual objects, yet lack a practical scheme for producing the complex vehicle interactions.

In contrast to the existing language interfaces, we propose a two-stage translation from vehicle interactions and behaviors, to trajectories. Real driving logic guides the entire translation process, resulting in a high realism of the generated vehicle trajectories. Additionally, we propose random locality attention that effectively utilizes information from vehicle interactions to guide the trajectory generation, thus enabling the generated trajectories to exhibit complex interactions.

## 3 TRAJECTORY-LLM

We present the pipeline of using Traj-LLM to conduct the "interaction-behavior-trajectory" translation. Figure 2 provides an overview of Traj-LLM, which is divided into two stages (see **Interaction-Behavior Translation** in Figure 2(a) and **Behavior-Trajectory Translation** in Figure 2(b)).

**Interaction-Behavior Translation**  At the first stage illustrated in Figure 2(a), Traj-LLM translates the text of vehicle interactions to behaviors. We denote the text of vehicle interactions as $\mathbf{I}$, which records $O$ pairs of vehicle-to-vehicle interaction between $N$ vehicles. Each pair of interactions is documented in text like "Car A overtakes Car B" and "Car B bypasses Car C". We keep these texts short to enhance the convenience of using Traj-LLM. We feed $K$ text segments of $\mathbf{I}$ into LLM $\mathcal{T}$, which outputs the text of the behaviors $\mathbf{B}$ for $N$ vehicles. We document the behaviors of each vehicle like "[Car A] Change Lane: Both the left and right sides have ample road widths...".

We feed the text interactions $\mathbf{I}$ into the text embedding layer $\mathcal{E}$ of LLM $\mathcal{T}$ to compute the interaction feature $\mathbf{F}_I \in \mathbb{R}^{O \times C}$ in Eq. (1). $C$ counts the feature channels. The map $\mathbf{M} \in \mathbb{R}^{E \times (D \times U)}$ in the traffic scenario is represented by $E$ polylines with $D$ points in each polyline, and $U$ denotes the number of attributes for each point. We input a map $\mathbf{M}$ and the initial trajectories $\mathbf{X} \in \mathbb{R}^{N \times (S \times 4)}$

of $N$ vehicles to a polyline encoder $\mathcal{P}$ in (Shi et al., 2022; Gao et al., 2020) to compute the initial trajectories $\mathbf{F}_X \in \mathbb{R}^{N \times C}$ and the map feature $\mathbf{F}_M \in \mathbb{R}^{E \times C}$ as:

$$\mathbf{F}_I = \mathcal{E}(\mathbf{I}), \quad \mathbf{F}_X = \mathcal{P}(\mathbf{X}), \quad \mathbf{F}_M = \mathcal{P}(\mathbf{M}). \tag{1}$$

Each initial trajectory includes $S$ waypoints, and each waypoint is associated with four motion parameters (i.e., $x/y$-coordinates, heading, and speed). $E$ denotes the number of polylines in the map $\mathbf{M}$. The initial trajectories and the map together represent the spatial configuration of vehicles.

To ensure the consistency between vehicle interaction and map for preventing invalid situations such as vehicles moving beyond the map boundaries, we propose a random locality attention between the interaction $\mathbf{I}$ (see Figure 3), the initial trajectories $\mathbf{X}$, and the specific map $\mathbf{M}$ as:

$$\mathbf{S}^o_{I \leftrightarrow X} = softmax \left( \frac{\mathbf{F}^o_I \cdot \mathbf{F}^1_X}{\sqrt{C}}, \ldots, \frac{\mathbf{F}^o_I \cdot \mathbf{F}^N_X}{\sqrt{C}} \right), \quad \left\{ \mathbf{S}^{o,1}_{I \leftrightarrow X}, \ldots, \mathbf{S}^{o,K}_{I \leftrightarrow X} \right\} = top_K \left( \mathbf{S}^{o,1}_{I \leftrightarrow X}, \ldots, \mathbf{S}^{o,N}_{I \leftrightarrow X} \right),$$

$$\mathbf{S}^o_{I \leftrightarrow M} = softmax \left( \frac{\mathbf{F}^o_I \cdot \mathbf{F}^1_M}{\sqrt{C}}, \ldots, \frac{\mathbf{F}^o_I \cdot \mathbf{F}^E_M}{\sqrt{C}} \right), \quad \left\{ \mathbf{S}^{o,1}_{I \leftrightarrow M}, \ldots, \mathbf{S}^{o,K}_{I \leftrightarrow M} \right\} = top_K \left( \mathbf{S}^{o,1}_{I \leftrightarrow M}, \ldots, \mathbf{S}^{o,E}_{I \leftrightarrow M} \right), \tag{2}$$

$\mathbf{S}_{I \leftrightarrow X} \in \mathbb{R}^{O \times N}$ and $\mathbf{S}_{I \leftrightarrow M} \in \mathbb{R}^{O \times E}$ represent the correlation scores between $O$ pairs of vehicle interaction, $N$ initial trajectories and $E$ polylines in the map $\mathbf{M}$. In Eq. (2), $top_K$ operation selects the top-K largest scores from the set of correlation scores. The selected scores are used to weight the initial trajectory feature $\mathbf{F}_X$ and map features $\mathbf{F}_M$, which are added to the feature $\mathbf{F}_I$ as:

$$\widehat{\mathbf{F}}^o_I = \mathbf{F}^o_I + \sum_{k=1}^K (\alpha^k_{I \leftrightarrow X} \mathbf{S}^{o,k}_{I \leftrightarrow X} \cdot \mathbf{F}^k_X + \alpha^k_{I \leftrightarrow M} \mathbf{S}^{o,k}_{I \leftrightarrow M} \cdot \mathbf{F}^k_M), \quad \alpha^k_{I \leftrightarrow X}, \alpha^k_{I \leftrightarrow M} \sim \mathcal{N}(0,1). \tag{3}$$

By setting $K < N$ and $E$, we let the feature $\widehat{\mathbf{F}}_I \in \mathbb{R}^{O \times C}$ capture the locality correlation between the interaction and the spatial configuration of the map. This locality enables each pair of vehicle interactions to occur at the most reasonable locations on the map. Besides, $\alpha^k_{I \leftrightarrow X}, \alpha^k_{I \leftrightarrow M}$ are random variables obeying the normal distribution, which jitters the correlation scores $\mathbf{S}^{o,k}_{I \leftrightarrow X}, \mathbf{S}^{o,k}_{I \leftrightarrow M} \in \mathbb{R}$. They let the feature $\widehat{\mathbf{F}}^o_I \in \mathbb{R}^C$ represent the $o^{th}$ interaction, whose location has some jitter within a range.

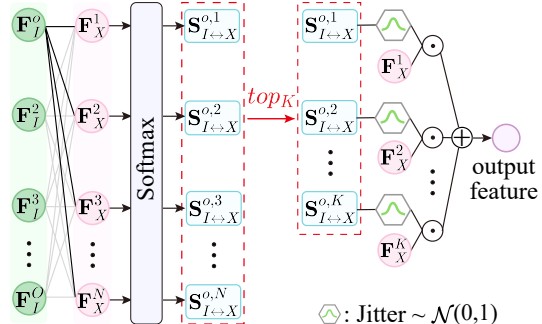

Figure 3: Illustration of the random locality attention. Here, we use the attention between $O$ interaction features $\{\mathbf{F}^o_I \mid o = 1, ..., O\}$ and $N$ initial trajectory features $\{\mathbf{F}^n_X \mid n = 1, ..., N\}$ as an example. This illustration applies to the attention formulated in Eqs. (2) and (6).

Given the interaction feature $\widehat{\mathbf{F}}_I$, LLM $\mathcal{T}$ uses the regression head to output $N$ text segments in $\mathbf{B}$. We denote this regression head as $\mathcal{H}(query, key, value)$, which has a cross-attention architecture with tunable parameters denoted as:

$$\mathbf{B} = \mathcal{H}(\overline{\mathbf{B}}, \sigma(\widehat{\mathbf{F}}_I), \sigma(\widehat{\mathbf{F}}_I)), \quad \mathbf{L}_B = \mathcal{L}_{CE}(\mathbf{B}, \widehat{\mathbf{B}}). \tag{4}$$

In Eq. (4), $\sigma$ means the fully-connected layers with non-linear activation. The regression head follows the autoregressive style, utilizing the interaction feature $\widehat{\mathbf{F}}_I$ and the already output behavioral text $\overline{\mathbf{B}}$ to update the final behavioral text $\mathbf{B}$. Here, the already output text $\overline{\mathbf{B}}$ queries the contextual information within the interaction feature $\widehat{\mathbf{F}}_I$, which helps to output vehicle behaviors aligned with the interactions. During the training phase, we use the cross-entropy loss $\mathbf{L}_B$ to penalize the difference between the regressed segments in $\mathbf{B}$ and the ground-truth segments in $\widehat{\mathbf{B}}$.

**Behavior-Trajectory Translation** At the second stage illustrated in Figure 2(b), Traj-LLM translates the textual description of vehicle behaviors in $\mathbf{B}$ to trajectories $\mathbf{T} \in \mathbb{R}^{N \times (S \times 4)}$ for $N$ vehicles. We convert numerical data of trajectories into text for training and inference of LLM. Again, each trajectory includes $S$ waypoints with four motion parameters.

Figure 2(b) illustrates the behavior-trajectory translation. We input the text segment of the behaviors $\mathbf{B}$ into LLM $\mathcal{T}$, and text embedding layer $\mathcal{E}$ computes the behavior feature $\mathbf{F}_B \in \mathbb{R}^{N \times C}$ as:

$$\mathbf{F}_B = \mathcal{E}(\mathbf{B}). \tag{5}$$

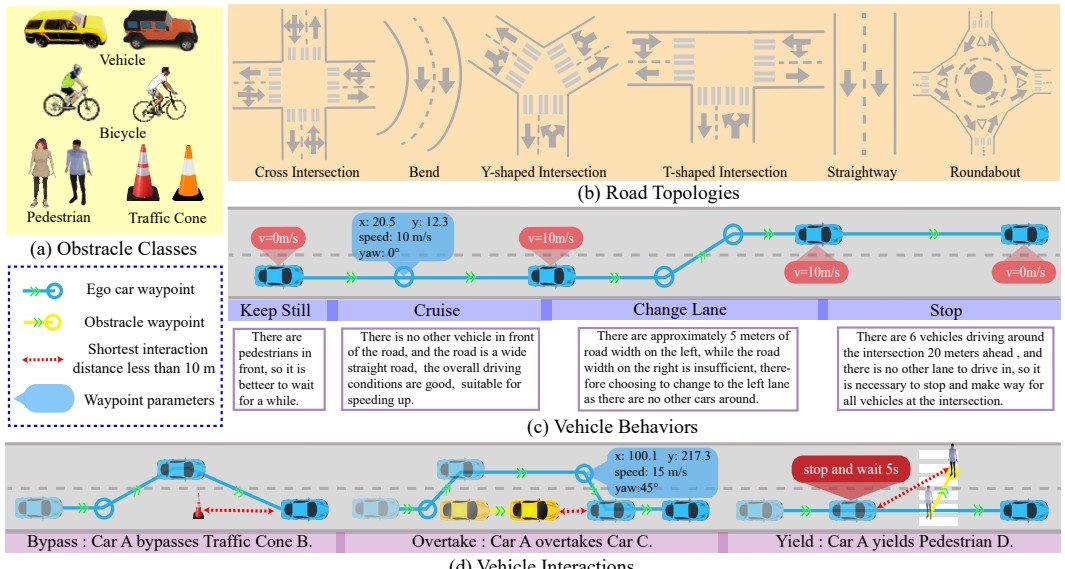

Figure 4: (a) We prepare four kinds of objects (i.e., vehicle, pedestrian, bicycle, and traffic cone). (b) There are six kinds of road topologies, where the road shapes are different. (c) Each object can take the behaviors of changing lanes, cruising, stopping, and keeping still. (d) Bypassing the static object, overtaking and yielding the dynamic object are the interactions in the L2T dataset.

We employ the random locality attention to capture the correlation between the behavior feature $\mathbf{F}_B$ and the interaction feature $\widehat{\mathbf{F}}_I$, computing the correlation scores $\mathbf{S}_{B\leftrightarrow I} \in \mathbb{R}^{N\times O}$ as:

$$\mathbf{S}_{B\leftrightarrow I}^n = softmax\left(\frac{\mathbf{F}_B^n\cdot\widehat{\mathbf{F}}_I^1}{\sqrt{C}},\ldots,\frac{\mathbf{F}_B^n\cdot\widehat{\mathbf{F}}_I^O}{\sqrt{C}}\right),\ \left\{\mathbf{S}_{B\leftrightarrow I}^{n,1},\ldots,\mathbf{S}_{B\leftrightarrow I}^{n,K}\right\} = top_K\left(\mathbf{S}_{B\leftrightarrow I}^{n,1},\ldots,\mathbf{S}_{B\leftrightarrow I}^{n,O}\right), \quad (6)$$

where $K < O$. We use the correlation scores $\mathbf{S}_{B\leftrightarrow I}$ to weight the interaction feature $\widehat{\mathbf{F}}_I$ as:

$$\widehat{\mathbf{F}}_B^n = \mathbf{F}_B^n + \sum_{k=1}^{K}\alpha_{B\leftrightarrow I}^k\mathbf{S}_{B\leftrightarrow I}^{n,k}\cdot\widehat{\mathbf{F}}_I^k, \quad \alpha_{B\leftrightarrow I}^k\sim\mathcal{N}(0,1), \quad (7)$$

where we sample the random variable $\alpha_{B\leftrightarrow I}^k$ from the normal distribution. In Eq. (7), we compute the behavior feature $\widehat{\mathbf{F}}_B^n \in \mathbb{R}^C$ for the $n^{th}$ vehicle. The random locality attention focuses the behavior feature $\widehat{\mathbf{F}}_B^n$ on the most relevant interaction features and the map information contained within them. The random variable $\alpha_{B\leftrightarrow I}^k$ also jitters the behavior feature $\widehat{\mathbf{F}}_B^n$, allowing Traj-LLM to generate more diverse trajectories aligned with the interactions and the map.

We feed the behavior feature $\widehat{\mathbf{F}}_B \in \mathbb{R}^{N\times C}$ of $N$ vehicles to the tunable head $\mathcal{H}(qurey, key, value)$ of LLM $\mathcal{T}$ to generate the trajectories in $\mathbf{T}$ as:

$$\mathbf{T} = \mathcal{H}(\overline{\mathbf{T}}, \sigma(\widehat{\mathbf{F}}_B), \sigma(\widehat{\mathbf{F}}_B)), \quad \mathbf{L}_T = \mathcal{L}_{CE}(\mathbf{T}, \widehat{\mathbf{T}}), \quad (8)$$

where $\overline{\mathbf{T}}$ is the already-outputted trajectories. It queries the behavior information from $\widehat{\mathbf{F}}_B$ during the generation of the trajectories in $\mathbf{T}$. We minimize the cross-entropy loss $\mathbf{L}_T$ to penalize the difference between the generated trajectories $\mathbf{T}$ and the ground-truth trajectories $\widehat{\mathbf{T}}$.

## 4 LANGUAGE-TO-TRAJECTORY DATASET

Considering the high risk of traffic accidents associated with collecting vehicle trajectories with high-intensity interactions on real roads, we opt to simulate the interactive trajectories in the virtual world. We construct a new simulator to increase the realism of vehicle trajectories collected in the simulated environment. The L2T dataset contains 240K road scenes, where the object classes, road topologies, and interactions are illustrated in Figure 10. We divide the L2T dataset into training and

testing sets, which contain 200K and 40K scenes, respectively. Below we briefly provide the basic information of the L2T dataset. More information can be found in the App A.

**Object Classes** Each object can be taken from the classes of vehicle, pedestrian, bicycle, and traffic cone (see Figure 10(a)), which are widely seen in reality. Expect traffic cones to always be static; other objects can be stationary or moving. These objects may be of different sizes in road scenes. In this paper, we only use the bird's-eye view projection of objects to determine their sizes.

**Road Topologies** We prepare six typical kinds of road topology, including straightway, bend, roundabout, cross/T-shaped/Y-shaped intersection (see Figure 10(b)). Each kind of road topology has variant shapes and lanes in the training and testing sets. A set of 2D coordinates represents the boundary of each lane. We use the XODR file to store each road topology with multiple lanes.

**Vehicle Behaviors** Each vehicle has the behaviors of changing lanes, cruising, keeping still, and stopping. Cruising or changing lanes means moving along the same lane or across lanes. We limit the velocity during vehicle cruising to 120km/h. Keeping still/stopping is temporary/permanent. As illustrated in Figure 10(c), we document a vehicle's behaviors sequentially. We recruit a group of drivers with rich experience to explain and document the logic behind each behavior.

**Vehicle Interactions** A vehicle may interact with 1~5 static/moving objects. In contrast to the behavior of a single object, the interaction defined here captures the relationship between a vehicle and an object. The possible interactions present in the L2T dataset are illustrated in Figure 10(d). The vehicle can bypass a static object to avoid collision. It can also overtake or yield the moving object. During overtaking or yielding, the vehicle occupies a lane sooner or later than the interactive object. We focus on the short-range interactions between objects. These high-intensity interactions should be completed in a short range of less than 10 meters, where the vehicle must take prompt action to avoid collision with the objects. The recruited drivers document each interaction in text. We employ a two-level check to evaluate the quality of annotations. Firstly, a rule-based script automatically examines whether the interactions match the annotated content. Secondly, the results of the automated check are randomly grouped, and a fixed-proportion random check is conducted by drivers. More details can be found in the App A.

**Vehicle Trajectories** Every moving object has a trajectory, which consists of a series of waypoints arranged chronologically. As illustrated in Figure 10(c–d), we associate each waypoint with four motion parameters (i.e., x/y-coordinates, heading, and speed). The duration of all the trajectories we provide is within 20 seconds and the sampling frequency is 20 Hz.

## 5 EXPERIMENTAL RESULTS

### 5.1 REALISM, DIVERSITY, AND CONTROLLABILITY

We use Llama-7B (Touvron et al., 2023) as the language model used in Traj-LLM, as it is among the most popular open-source large language models and demonstrates superior performance in understanding traffic scenarios. For more implementation details of Traj-LLM, see App C. Below, we evaluate the realism and diversity of the trajectories generated by various methods. We also compare the control capability of different methods for trajectories to verify if these methods can generate the expected trajectories according to the interactions specified in the text.

The compared methods differ in their control conditions for generating vehicle trajectories. For a fair comparison, all methods utilize the L2T for training and testing. SNet (Bergamini et al., 2021) and TSim (Suo et al., 2021) use the BEV road topologies to generate trajectories. We train BITS (Xu et al., 2023) and CTG (Zhong et al., 2023b) on the trajectories from L2T, relying on rule-based conditions for trajectory generation as in (Xu et al., 2023; Zhong et al., 2023b). CTG++ (Zhong et al., 2023a), LGen (Tan et al., 2023), and Traj-LLM utilize text of interaction as the condition.

**Realism** In Table 1 "Realism", we report the agent- and scene-level (Zhong et al., 2023b;a) realism of the generated trajectories. The agent-level realism measures the distribution distances of longitudinal acceleration magnitude (**LO**), lateral acceleration magnitude (**LA**), jerk (**JE**), and yaw rate (**YR**), and their average (**AVG**), between each agent's generated and ground-truth trajectories. The scene-level realism is assessed by the relative scores of **LO**, **LA**, **JE**, **YR**, and **AVG**. More details about calculating these metrics can be found in App B. To compute a relative score, we gauge the distribution distance between each pair of vehicles' generated/ground-truth trajectories. Then, we evaluate the difference between the above two distances as the relative score.

Table 1: We compare Traj-LLM with state-of-the-art methods on the L2T dataset. The results measure the realism and diversity of the trajectories generated by various methods. The realism score has been multiplied by 100. ↓/↑ means a smaller/larger value represents a better performance.

| Method | | Realism | | | | | | | | | | Diversity | | | |
| | | Agent | | | | | Scene | | | | | Agent | | | Scene |
| | | LO↓ | LA↓ | JE↓ | YR↓ | AVG↓ | LO↓ | LA↓ | JE↓ | YR↓ | AVG↓ | MD↑ | AD↑ | FD↑ | WD↑ |
|---|---|---|---|---|---|---|---|---|---|---|---|---|---|---|---|
| Image | SNet | 14.20 | 9.84 | 8.38 | 7.92 | 10.09 | 11.28 | 4.78 | 2.76 | 8.72 | 6.88 | 0.00 | 0.00 | 0.00 | 0.00 |
| | TSim | 25.36 | 21.78 | 5.33 | 3.65 | 14.03 | 15.94 | 4.60 | 10.67 | 7.91 | 9.78 | 0.52 | 0.25 | 0.30 | 2.92 |
| Rule | BITS | 10.16 | 5.82 | 9.23 | 4.30 | 7.38 | 10.57 | **1.70** | 3.54 | 4.75 | 5.14 | 6.80 | 3.22 | 6.98 | 7.27 |
| | CTG | 8.26 | 7.33 | 19.92 | 1.68 | 9.30 | 10.40 | 6.14 | 10.10 | 3.64 | 7.57 | 6.49 | 3.44 | 6.03 | 5.70 |
| Text | LGen | 7.55 | 9.73 | 13.09 | 1.91 | 8.07 | 9.59 | 7.89 | 10.21 | 4.10 | 7.95 | 3.38 | 1.74 | 2.19 | 4.02 |
| | CTG++ | 6.13 | 7.13 | 21.19 | 1.13 | 8.90 | 7.94 | 4.16 | 10.37 | 2.22 | 6.17 | 6.38 | 3.55 | 6.08 | 5.79 |
| | **Traj-LLM** | **2.56** | **3.17** | **0.74** | **0.28** | **1.69** | **0.73** | 2.68 | **0.52** | **0.54** | **1.12** | **14.91** | **6.81** | **8.71** | **13.81** |

Traj-LLM surpasses other methods in terms of realism at both the agent and scene levels. Traj-LLM introduces the interaction-behavior translation. Through learning from the real data, the short text of the vehicle interactions is translated into the behaviors with driving logic. It makes each vehicle's trajectory consistent with the human driving habits, generating more natural trajectories. Since each scene contains multiple vehicles, the entire scene is also made more realistic.

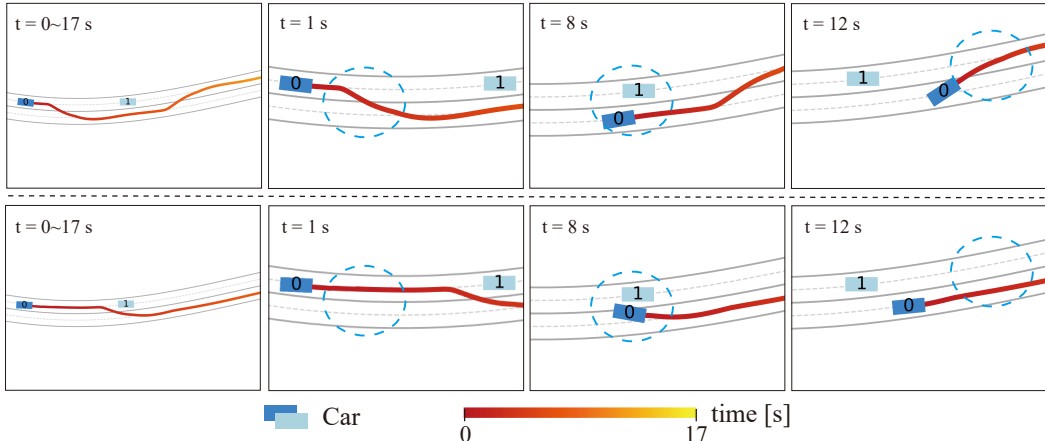

Figure 5: Diverse trajectories generated by Traj-LLM from the same initial positions.

**Diversity** In Table 1 "Diversity", we measure the agent-level diversity (Zhang et al., 2022) by reporting the map-aware average self distance (**MD**), average self distance (**AD**), and final self distance (**FD**). We measure the scene-level diversity (Xu et al., 2023) by reporting Wasserstein distance (**WD**). To evaluate the agent-level diversity of trajectories, we conduct multiple experiments with the same initial conditions (history trajectories, rules, images) and evaluate metrics for the same vehicle in every two experiments. **MD/AD** denotes the average L2 distance between the most divergent/similar trajectories. **FD** assesses the average L2 distance among the final position of the most similar trajectories. To evaluate the scene-level diversity, we compute the generated trajectory distribution on the map. Then, we calculate the density histogram of the distribution as the density profile. We measure the average **WD** between pairwise density profiles across multiple experiments.

Traj-LLM achieves better trajectory diversity, as it employs the random locality attention to the behavior-trajectory translation. It dynamically focuses on the road topology, the driving status of neighbor vehicles, and the history trajectories. We jitter the parameters of the driving trajectories, further enriching the diversity of generated trajectories. We visualize the generated trajectories in Figure 5, demonstrating that Traj-LLM generates more diverse trajectories.

**Controllability** In Table 2, we compare Traj-LLM with state-of-the-art methods that also rely on text to generate interactive vehicle trajectories, in terms of the scene-level controllability. We conduct multiple experiments for each vehicle and record the success rate of achieving the specified

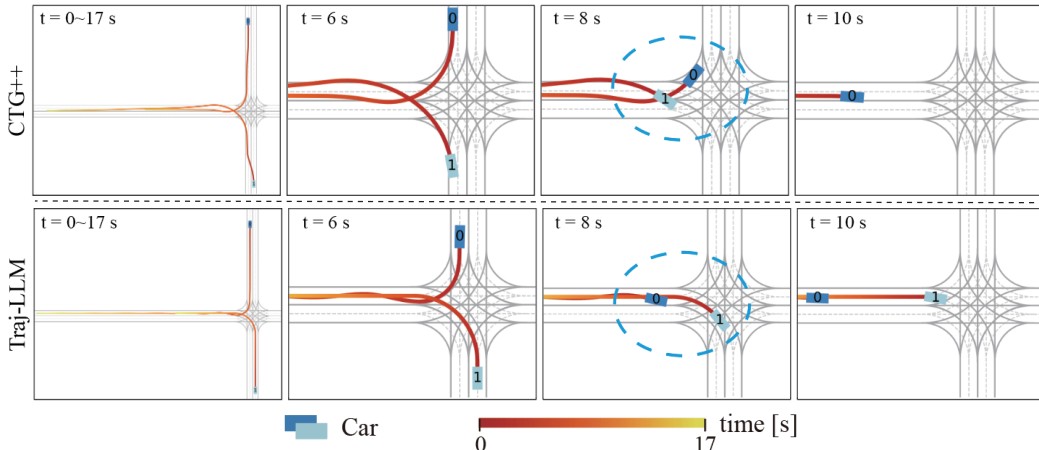

Figure 6: Controllability of CTG++ and Traj-LLM. In this example, Traj-LLM successfully control Car 0 to overtake Car 1, while the state-of-the-art CTG++ fails in this case.

interaction and adhering to traffic rules. We calculate the average success rate of different vehicles having various interactions over multiple experiments.

Traj-LLM achieves a higher success rate because it introduces the two-stage translation that generates realistic driving logic. This translation incorporates the random locality attention to enhance the realism and diversity of the generated trajectories. Solely relying on the short text, Traj-LLM presents a stronger control over the interactions than other methods (see Figure 6). Traj-LLM hopefully shows stronger controllability with more detailed control instructions.

Table 2: Comparison of interaction success rates(%) with text-based generation methods.

| Method | CTG++ | LCTGen | Traj-LLM |
|---|---|---|---|
| Overtake | 65.2 | 36.9 | **80.1** |
| Bypass | **91.8** | 54.5 | 81.3 |
| Yield | 77.7 | 45.5 | **83.5** |
| Avg | 78.2 | 45.6 | **81.6** |

## 5.2 ABLATION STUDY

In Table 3, we conduct the ablation study to measure the realism and diversity of the trajectories and the controllability of LLM. We denote **I-T/I-B-T** as "Interaction-Trajectory"/"Interaction-Behavior-Trajectory" translation. We use the **AVG**, **WD** and **success rate** for measuring the scene-level realism, diversity, and controllability. Without the driving behaviors that reflect the inherent logic of human driving, **I-T** produces inferior results (see the first row of Table 3) compared to **I-B-T**.

Table 3: **I-T/I-B-T** means "Interaction-Trajectory"/"Interaction-Behavior-Trajectory" translation. The results measure the realism, diversity and controllability (%).

| Translation | Logic | Random | Locality | Realism↓ | Diversity↑ | Control↑ |
|---|---|---|---|---|---|---|
| **I-T** | | | | 14.52 | 14.39 | 36.7 |
| **I-B-T** | | ✓ | ✓ | 3.08 | **22.64** | 48.2 |
| | ✓ | | | 4.81 | 9.66 | 62.1 |
| | ✓ | ✓ | | 1.45 | 17.26 | 58.3 |
| | ✓ | | ✓ | 2.60 | 12.55 | 67.9 |
| | ✓ | ✓ | ✓ | **1.12** | 13.81 | **81.6** |

In Table 3 "**I-B-T**", we list the critical modules of **I-B-T** translation, i.e., the driving-logic learning (**Logic**), random jitter (**Random**), and locality attention (**Locality**). Without **Logic** in the second row, we use LLM to generate the temporal sequence of behaviors only for each vehicle. This degrades realism, diversity, and controllability, demonstrating the importance of driving-logic learning. Based on **Logic**, we remove **Random** or **Locality** (see the third to fifth rows). Especially,

the alternative without **Random** and **Locality** degrades the random locality attention to a vanilla cross-attention (the third row). Compared to these alternatives, the complete Traj-LLM shows better performance. This is because **Locality** enables the vehicles to generate reasonable trajectories aligned with the interactions and the map, while **Random** makes the generated trajectories more diverse. Note that **Locality** also offers efficient constraints and affects diversity. The fifth row removes **Random** but diversity doesn't drop much compared with the last row, while in the third and fourth rows, when **Locality** is removed, **Random** significantly impacts diversity.

## 5.3 GENERATED DATA FOR TRAJECTORY PREDICTION

In Table 4, we evaluate the effectiveness of Traj-LLM, by using the generated vehicle trajectories to train downstream trajectory prediction models, i.e., MTR (Shi et al., 2022) and HDGT (Jia et al., 2023). Apart from the model training and validating on the same dataset, we follow the protocol of the cross-dataset validation in (Torralba & Efros, 2011), where we train and test each prediction model on different datasets. The cross-dataset validation helps to evaluate the robustness of the prediction model trained and tested on various data distributions. We utilize two parts of trajectory data from the training set of WOMD (Ettinger et al., 2021) (**WOMD (train)**) and the trajectories generated by Traj-LLM (**Traj-LLM (L2T)**), where Traj-LLM is trained on 200K scenarios of the L2T dataset. Each part of the training data contains 400K scenes. Each scene includes 2-8 vehicles with various interactions. The trained models are evaluated on the validation sets of WOMD (**WOMD (val)**) and L2T (**L2T (val)**), each containing about 40K scenes. We report the trajectory prediction results using **mAP**, **minADE**, and **Miss Rate**, as defined in WOMD.

Table 4: Results of trajectory prediction.

| Train | | Test (mAP ↑, minADE ↓, Miss Rate ↓) | | | |
|---|---|---|---|---|---|
| WOMD (train) | Traj-LLM (L2T) | WOMD (val) | | L2T (val) | |
| | | MTR | HDGT | MTR | HDGT |
| ✓ | | 0.405, 0.650, 0.141 | 0.272, 0.604, 0.151 | 0.211, 0.921, 0.267 | 0.134, 1.674, 0.491 |
| | ✓ | 0.106, 1.583, 0.459 | 0.149, 1.290, 0.434 | 0.340, 0.662, 0.154 | 0.196, 0.968, 0.280 |
| ✓ | ✓ | **0.416, 0.633, 0.139** | **0.294, 0.578, 0.148** | **0.384, 0.620, 0.145** | **0.243, 0.775, 0.194** |

By training MTR/HDGT on **Traj-LLM (L2T)** and validating it across on **WOMD (val)** (see blue cells in the second row), we find a performance degradation compared to the prediction models trained on **WOMD (Train)** (see the blue cells in the first row). This is because different trajectory distributions exit in **WOMD (Train)** and the generated data of **Traj-LLM (L2T)**. A large portion of the trajectories in **Traj-LLM (L2T)** represent the dense interactions between vehicles, whereas **WOMD (Train)** provides many vehicles without interaction. We also find a similar degradation in the purple cells of the first and second rows when training the prediction models on **WOMD (Train)** and validating it on **L2T (test)**.

By integrating the data of **WOMD (Train)** and the generated trajectories in **Traj-LLM (L2T)**, MTR and HDGT outperform (see the last row of Table 4) their counterparts trained on a single dataset. The quantitative and qualitative results demonstrate that the trajectories generated by Traj-LLM exhibit remarkable realism and diversity, thus benefiting downstream prediction models. Furthermore, we conduct ablation study of generated data and L2T under fixed / non-fixed total dataset size in App D.

## 6 CONCLUSION

The precision prediction of vehicle trajectories holds significance in autonomous driving. To train accurate trajectory prediction models, a substantial quantity of data of vehicle trajectories is imperative. The recent development of generative artificial intelligence makes it possible to produce a large amount of trajectory data for training the trajectory prediction models at the cost of less human effort. This paper proposes a trajectory generator, Traj-LLM, which relies on the short text description of vehicle interaction to produce the interactive trajectories efficiently. We utilize the large language model to build Traj-LLM. In contrast to the language-based generators for "interaction-trajectory" translation, Traj-LLM conducts the "interaction-behaviour-trajectory", where it possesses a powerful capability in understanding the interactions between vehicles and outputting their behaviors and driving logics. This capability allows Traj-LLM to produce realistic and diverse trajectory data for training downstream trajectory prediction models. We also contribute the L2T dataset, which contains the interactive trajectories associated with the text descriptions of vehicle interactions, behaviors, and driving logic, to train Traj-LLM and promote relevant research in the future.

ACKNOWLEDGMENTS

We thank the anonymous reviewers for their constructive suggestions. This work is supported by the National Natural Science Foundation of China (No. 62476192), Natural Science Foundation of Tianjin (No. 23JCQNJC02010), the National Research Foundation, Singapore and Infocomm Media Development Authority under its Trust Tech Funding Initiative (No. DTC-RGC-04). Any opinions, findings and conclusions or recommendations expressed in this material are those of the authors and do not reflect the views of National Research Foundation, Singapore and Infocomm Media Development Authority.

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
