## A  DETAILS OF L2T

In this section, we provide details of the L2T dataset. Each scenario in L2T is organized into individual JSON format files containing the following components.

**Interaction Section**  The interaction section is a summary of the map topology, the number of the agents, and their interactions. The interaction section of the example below serves as the context for understanding the interaction label.

**Behavior Section**  In the behavior section, we provide the initial position $(x, y)$, heading, and speed of each agent. Each agent is associated with a sequence of behaviors and their logic descriptions. Each behavior has a tag of KEEPSTILL, CRUISE, STOP, or CHANGELANE.

**Trajectory Section**  In the trajectory section, we provide the sequence of motion parameters for each agent for representing its trajectory. The motion parameters are $x/y$-coordinates, $x/y$-accelerations, yaw, and rest time. We organize the sequence of motion parameters in a frame-by-frame fashion.

```
{
  "scenario_label": {
      "map": "straight road",
      "interaction": "egocar0 bypasses npccar3821",
      "agent_nums": 2
  },
  "agent_behavior": {
      "egocar0": {
          "init": {
              "x": 401.938,
              "y": -303.069,
              "heading": 1.7837963267948966,
              "speed": 0
          },
          "behaviors": [
              {
                  "behavior": "KEEPSTILL",
                  "description": "wait for 1 seconds before the scenario start"
              },
              {
                  "behavior": "CRUISE",
                  "description": "To avoid collision, the egocar bypasses the npccar3821 from
                      the left side."
              }
          ]
      },
      "npccar3821": {
          "init": {
              "x": 394.742,
              "y": -253.616,
              "heading": 2.0107963267948965,
              "speed": 0
          },
          "behaviors": [
              {
                  "behavior": "CRUISE",
                  "description": "The npccar3821 is driving straight ahead."
              }
          ]
      }
  },
  "agent_trajectory": {
      "frame0": {
          "egocar0": {
              "agenttype": "AgentType.EGO",
              "position_x": 401.938,
              "position_y": -303.069,
```

```
            "heading": 1.7837963267948966,
            "speed": 0.0
        },
        "npccar3821": {
            "agenttype": "AgentType.NPC",
            "position_x": 394.742,
            "position_y": -253.616,
            "heading": 2.0107963267948965,
            "speed": 0.0
        }
    },
    "frame1": {
        "egocar0": {
            "agenttype": "AgentType.EGO",
            "position_x": 401.93798828125,
            "position_y": -303.069000244141,
            "heading": 1.7837963267948966,
            "speed": 0.0
        },
        "npccar3821": {
            "agenttype": "AgentType.NPC",
            "position_x": 394.742004394531,
            "position_y": -253.615997314453,
            "heading": 2.0107963267948965,
            "speed": 0.0
        }
    },
    ...
  }
}
```

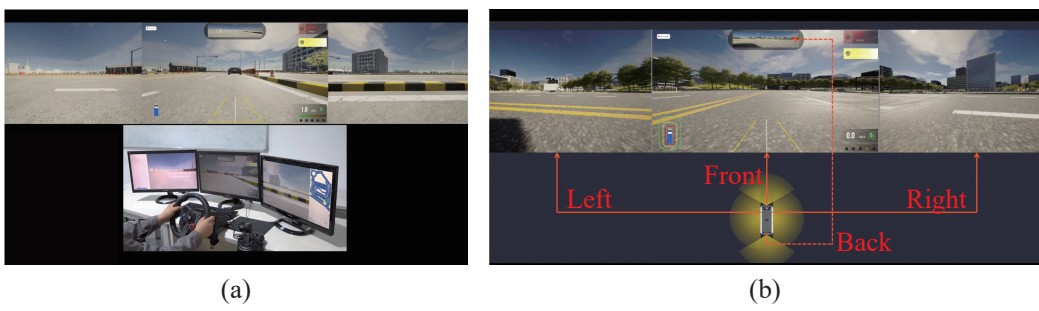

          (a)                        (b)

Figure 7: (a) The hardware component of our simulator. (b) The first-person view of the virtual road environment.

## A.1 DETAILS ABOUT THE COLLECTION AND ANNOTATION OF L2T

Our simulator integrates the re-developed hardware and software to efficiently collect and annotate the vehicle trajectories in the L2T dataset.

The hardware component of our simulator mainly includes a steering wheel, a gear shifter, gas and brake pedals, and several displays, as shown in Figure 7(a) of the supplementary file. The displays visualize the first-person view of the road environments from the left, right, front, and back sides of the car, as shown in Figure 7(b). Here, the road environments are simulated by LGSVL, a simulation engine for autonomous driving. Our hardware simulates a realistic car cockpit, helping users to drive on the virtual road to produce vehicle trajectories as real as possible.

The software component is mainly built on top of the LGSVL engine. LGSVL can record all trajectories of vehicles on the virtual road. We re-develop this engine to enable the connection among multiple sets of the above hardware, allowing multiple users to control different cars on the same

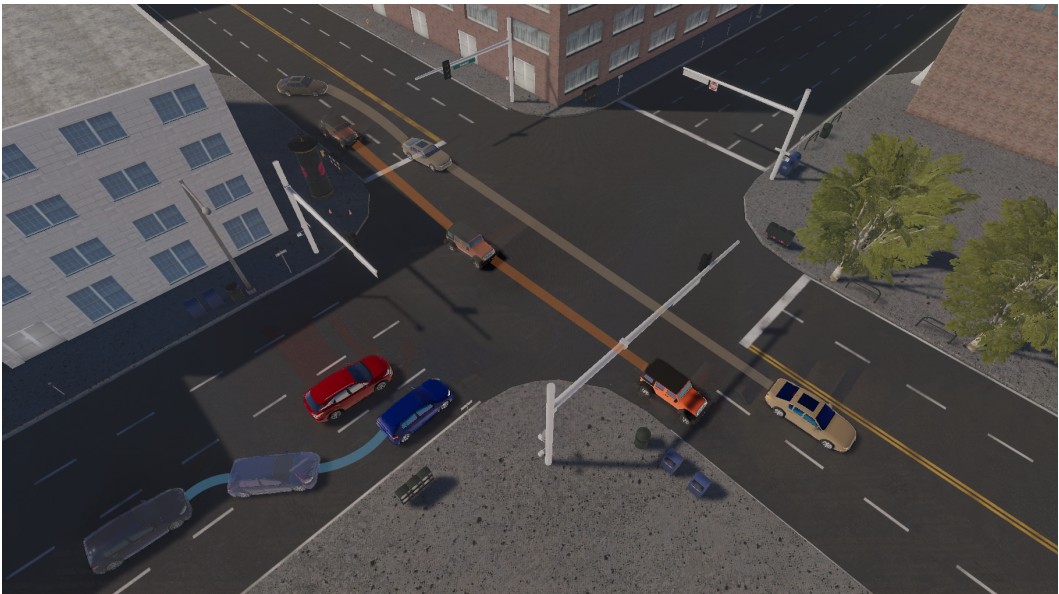

Figure 8: Multiple users control different cars on the same virtual road.

virtual road. This software facilitates the simulation of vehicle interactions like those in the real world. In Figure 8, we show the interactive cars in the virtual road from the third-person view.

We select drivers as the primary participants to create the L2T dataset. Based on the manually designed map, the drivers obtain trajectories by operating the above simulators. We recruit 30 drivers with rich experience to explain and document vehicle interactions and behaviors with driving logic. Here, we conduct a two-level check to evaluate the annotation quality. We develop an automated Python script at the first check level to verify whether the trajectories align with the text interactions and behaviors. These scripts incorporate some basic checking rules, such as the rule that the overtaking between two vehicles must involve acceleration, and the behavior parameters within the acceleration must be equivalent to the motion parameters of the trajectories.

At the second check level, we divide the scenarios that pass the first check level into 10 groups. Three drivers check the annotations of each group of scenarios. We randomly select 20% of scenarios from each group. If half of the selected scenarios have false annotations, all scenarios in this group must be re-annotated; otherwise, only the scenarios with false annotations are re-annotated.

Since our work concerns the driving logic of drivers in the real world, the trajectories collected through our method closely resemble the real-world data.

### A.2 MAPS AND DATA DISTRIBUTIONS

The L2T dataset contains six kinds of road topologies, including straightway, bend, roundabout, cross/T-shaped/Y-shaped intersection. We provide the examples of maps in Figure 9. In Figure 10(a), we report the proportion of each type of map. Due to the complexity of roundabout maps and the high cost of annotations required, the proportion of roundabout data in the dataset is relatively low. The proportion of interactions and combinations under each type of map is relatively uniform.

In Figure 10(b), we further report the proportion of different types of behaviors that occur along with each type of interactions and combinations. Since most vehicles in the dataset are in driving mode at most of the time, the proportion of cruising is the highest among the optional behaviors.

### A.3 COMPARISONS BETWEEN L2T AND OHTER DATASETS

To validate the characteristics of L2T in comparison to real-world datasets, we conduct an in-depth comparison between L2T and other datasets like Waymo. Figure 11 illustrates the similarity be-

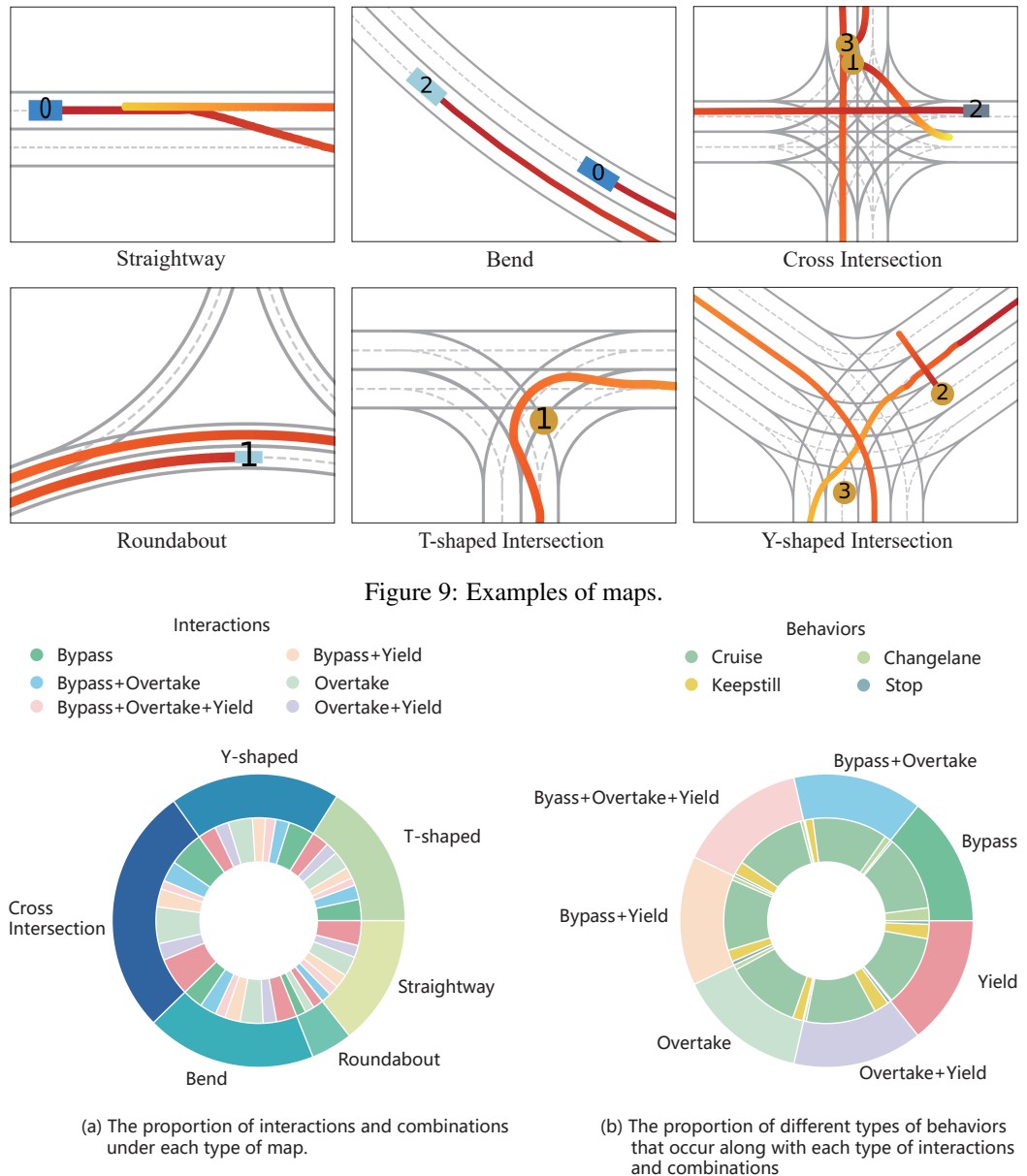

Figure 9: Examples of maps.

Figure 10: Statistics of the L2T dataset.

(a) The proportion of interactions and combinations under each type of map.

(b) The proportion of different types of behaviors that occur along with each type of interactions and combinations

tween L2T and other datasets regarding vehicle trajectory shapes and the number of objects of interest within the scene. It should be noted that due to the removal of traffic cones in the WOMD dataset. L2T retains them as stationary objects, L2T exhibits a higher number of stationary in overall trajectory shapes.

In Figure 12, we have compared the data distribution of our L2T and other datasets with trajectories in the real world. Specifically, we compare the distributions of trajectories' shapes represented by the waypoints' coordinates. The distributions are similar, showing that the vehicle trajectories in the L2T dataset are realistic, like those in the Argoverse and WOMD datasets.

## A.4 CORE IMPACT IN L2T AND TRAJ-LLM

Our method can generate strong interaction trajectory data similar to real scenes through interactive text and can improve real data's performance, proving our effectiveness. To better quantify it, we propose a novel metric: the scene-level closest interaction distance (CID), which is defined as the

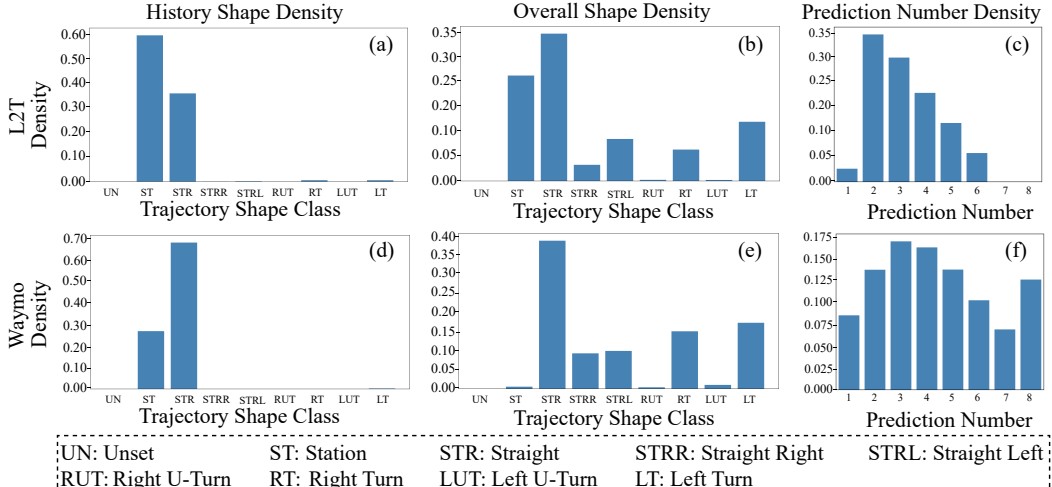

UN: Unset      ST: Station      STR: Straight      STRR: Straight Right      STRL: Straight Left

RUT: Right U-Turn      RT: Right Turn      LUT: Left U-Turn      LT: Left Turn

Figure 11: Comparison of the L2T and WOMD datasets in terms of trajectory shapes and prediction numbers.

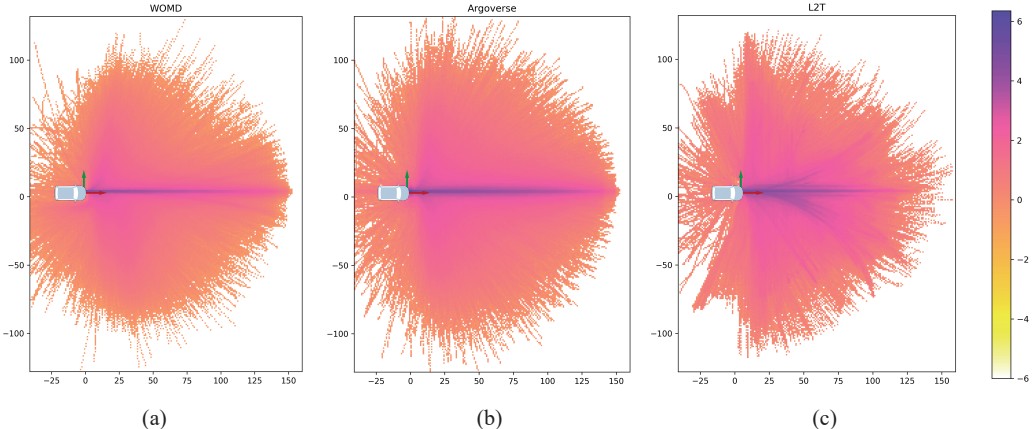

Figure 12: Comparison of trajectories' shape distributions of (a) WOMD, (b) Argoverse, and (c) L2T datasets.

average of the closest distances between the ego and other vehicles/obstacles in the current scene during the entire movement process. Figure 13 shows the density map (green and red regions) of the CID in the L2T dataset and the WOMD dataset. It can be seen that the closest interaction distance of our data is closer and the interactivity is stronger. Meanwhile, the scatter plot shows that after adding our data, the improvement of WOMD's performance is mainly concentrated in the interaction scenes with closer distances, further validating the effectiveness of our method.

## B    EVALUATION METRICS

We use the metrics below to evaluate the realism, diversity, and controllability.

**Realism**    To evaluate the realism of the generated trajectories, we compare them with the ground truth trajectories. This comparison is computed via the Wasserstein distance between the normalized histograms of the driving profiles for the generated and ground truth trajectories as:

$$\mathbf{Realism}_X = \mathcal{W}(\mathcal{N}orm(Hist(\mathbf{X}_{gen})), \mathcal{N}orm(Hist(\mathbf{X}_{gt}))). \tag{9}$$

In Eq (9), the symbol $\mathcal{W}$ represents the Wasserstein distance, while $X$ is the driving profile being compared. Realism is measured at both the agent and scene levels. Agent-level realism involves comparing four driving profiles: longitudinal acceleration magnitude (**LO**), latitudinal acceleration magnitude (**LA**), jerk (**JE**), and yaw rate (**YR**).

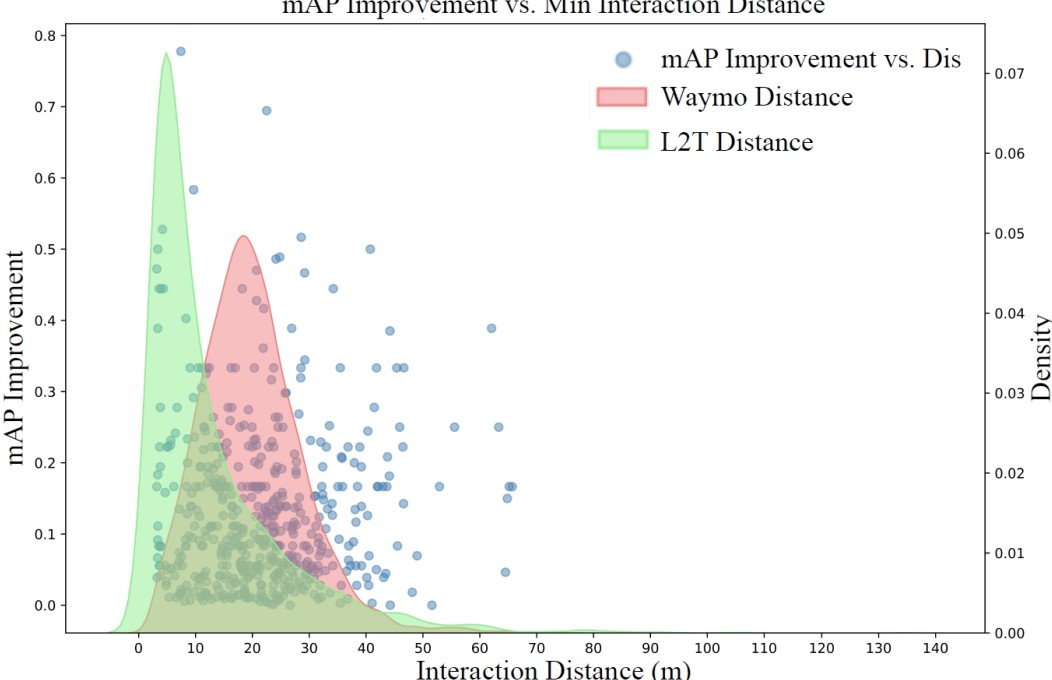

Figure 13: The distribution of interaction distances and the improvement in mAP when adding L2Tdata into the training set for MTR, as evaluated on the WOMD (val).

Scene-level realism, on the other hand, compares four relative driving profiles between agents: relative longitudinal acceleration magnitude, relative lateral acceleration magnitude, relative jerk, and relative yaw rate.

**Diversity**  We evaluate the agent- and scene-level diversity. We use the average self distance(AD), final self distance(FD), and map-aware average self distance (MD) to measure agent-level diversity of trajectories generated multiple times.

In Eq. (10), **AD** denotes the average L2 distance of all waypoint positions between the two closest generated trajectories for each agent.

$$\mathbf{AD} = \frac{1}{NDS} \sum_{n=1}^{N} \sum_{d=1}^{D} \min_{d' \in \mathcal{D} \wedge d' \neq d} \sum_{s=1}^{S} \left\| \mathbf{T}_s^{n,d} - \mathbf{T}_s^{n,d'} \right\|^2 . \tag{10}$$

In Eq. (11), **FD** denotes the average L2 distance of the final position between the two closest generated trajectories for each agent.

$$\mathbf{FD} = \frac{1}{ND} \sum_{n=1}^{N} \sum_{d=1}^{D} \min_{d' \in \mathcal{D} \wedge d' \neq d} \left\| \mathbf{T}_S^{n,d} - \mathbf{T}_S^{n,d'} \right\|^2 . \tag{11}$$

In Eq. (12), **MD** denotes the average L2 distance of all agents' trajectories between the two most distinct generated scenarios.

$$\mathbf{MD} = \max_{d,d' \in \mathcal{D}} \frac{1}{NS} \sum_{n=1}^{N} \sum_{s=1}^{S} \left\| \mathbf{T}_s^{n,d} - \mathbf{T}_s^{n,d'} \right\|^2 . \tag{12}$$

To measure scene-level diversity, we use Kernel Density Estimation to calculate trajectory density profile $\rho$ for each sample scenario. We calculate the average Wasserstein distance (**WD** in Eq. (13)) of density profiles for pairwise sample scenarios, as the scene-level diversity score.

$$\mathbf{WD} = \frac{2}{D(D-1)} \sum_{i=1}^{D-1} \sum_{j=i+1}^{D} \mathrm{W}\left(\rho_i, \rho_j\right) . \tag{13}$$

In Eqs. (10)-(13), $N$ is the number of agents in each traffic scenario. $\mathcal{D}$ is the set of generated samples for each scenario and $D$ is the number of samples in $\mathcal{D}$. $S$ is the length of generated trajectories.

**Controllability** For text-controlled methods, we evaluate the controllability of generated trajectories via the success rate (**SR** in Eq. (14)) of achieving text-specified interaction (overtake, bypass, and yield), which is the portion of interactive samples ($IS$) to total samples ($TS$).

$$\textbf{SR} = \frac{IS}{TS}. \tag{14}$$

### B.1 The calculation of controllability

We utilize a rule-based automatic behavior detection method to calculate **SR**. This method determines behaviors (such as overtake, bypass, yield, etc.) based on the interaction distance, speed, trajectory shape, etc. of the vehicle during movement, and assigns a confidence level reflecting the certainty of the results. Notably, we apply the same judging method for all methods, and only incorporate high-confidence results in our controllability calculations.

## C Implementation Details of Traj-LLM

We train Llama with $8\times$A800 GPUs. We utilize Low-Rank Adaptation (Hu et al., 2022) (LoRA) with a rank of 8 to fine-tune Llama. The LoRA alpha value is set to 32 to control the relative strength of the update compared to the original model's weights. A dropout rate of 0.1 is applied in LoRA to prevent overfitting. The training process span two epochs with a batch size of 16 for each iteration. We set the learning rate to 2e-5. The training data comprises 200K samples from the L2T training set. The entire training takes about 14 hours.

## D Ablation Study of Generated Data and L2T

In Table 5, we combine the original data, i.e. L2T, with the training data to train downstream trajectory prediction models MTR(Shi et al., 2022).

By adding all the data from **WOMD (Train)**, the trajectories of **L2T (train)** and the generated trajectories in **Traj-LLM (L2T)** (the last row), MTR performs better than without **L2T (train)** (the fourth row) or **Traj-LLM (L2T)** (the first row). This is because the L2T dataset is collected from real-world scenarios, which reflects the complexity and diversity of the real world. The real-world nature of the data enables the trajectory prediction models to learn from the real-world variations. Furthermore, Traj-LLM can generate numerous trajectories to enrich the training data. By combining the original data in **L2T (train)** with the generated trajectories in **Traj-LLM (L2T)**, we leverage the strengths of both, enhancing the accuracy and robustness of the trajectory prediction models.

Given the training data from **WOMD (Train)**, MTR achieves a better performance when it is trained on the original trajectories in **L2T (train)** and the generated trajectories in **Traj-LLM (L2T)** (the last row), compared to the training on the generated trajectories only (the fourth row). Though the generated trajectories may be diverse, they may still fail to capture the real-world trajectories' natural pattern fully. Combining generated data with original data can achieve a better balance in the training set. We find that this improvement is marginal. To some extent, this demonstrates the high quality of the generated trajectories that serve as training examples to strengthen the model's generalization of unseen scenarios.

In the second, third, and fourth rows of Table 5, the performance of MTR improves with increasing amounts of generated trajectories. This enhancement is attributed to using richer trajectory data from the Traj-LLM to train the trajectory prediction model.

In order to separate the utility of Traj-LLM data from just the additional data volume, we conduct another version of Table 5 with fixed total dataset size, the results are reported in Table 6. we define 100% as the same size as WOMD(train). It can be seen that the performance on the WOMD(val) drops accordingly as the proportion of WOMD (train) decreases. This is because WOMD and Traj-LLM (L2T) have different distributions (see A.3,A.4). Specifically, Traj-LLM (L2T) focuses

Table 5: Results of trajectory prediction on MTR.

| Train | | | Test (mAP ↑, minADE ↓, MissRate ↓) |
|---|---|---|---|
| WOMD (train) | L2T (train) | Traj-LLM (L2T) | WOMD (val) |
| 100% | 100% | 0 | 0.410, 0.977, 0.173 |
| 100% | 0 | 30% | 0.397, 1.030, 0.207 |
| 100% | 0 | 50% | 0.412, 0.903, 0.164 |
| 100% | 0 | 100% | 0.416, 0.633, **0.139** |
| 100% | 100% | 100% | **0.417, 0.632**, 0.152 |

on strong interaction scenarios, whereas WOMD distributes on weak interaction scenarios. When combining Traj-LLM (L2T) and WOMD (train) as the training set, the test set WOMD (val) has a different distribution from the training set. This will lead to a decrease in test performance. Furthermore, the generated data, Traj-LLM (L2T), can improve the trajectory prediction performance in strong-interaction scenarios (see A.4), as evidenced in Table 4, thereby demonstrating the effectiveness of our dataset. Table 7 shows that QCNet (Zhou et al., 2023), which achieves the state-of-the-art results on the Argoverse dataset, can be further improved by the generated trajectories. In Table 8, we compare Traj-LLM with state-of-the-art methods on the WOMD dataset in terms of the realism and diversity of the generated trajectories.

Table 6: Results of trajectory prediction on MTR under fixed total dataset size.

| Train | | Test (mAP ↑, minADE ↓, MissRate ↓) |
|---|---|---|
| WOMD (train) | Traj-LLM (L2T) | WOMD (val) |
| 100% | 0 | 0.405, 0.650, 0.141 |
| 50% | 50% | 0.383, 0.738, 0.196 |
| 30% | 70% | 0.353, 1.378, 0.249 |

Table 7: Results of trajectory prediction on the Argoverse dataset.

| Train | | Test ($minFDE_6$ ↓, $minADE_6$ ↓, $MissRate_6$ ↓) | |
|---|---|---|---|
| Argoverse (train) | Traj-LLM(L2T) | Argoverse (val) | L2T (val) |
| ✓ | | 1.243, 0.717, 0.157 | 1.795, 0.933, 0.251 |
| | ✓ | 2.704, 1.261, 0.379 | 1.490, 0.834, 0.143 |
| ✓ | ✓ | **1.231, 0.710, 0.149** | **1.452, 0.815, 0.136** |

Table 8: We compare Traj-LLM with state-of-the-art methods on the WOMD dataset. The results measure the realism and diversity of the trajectories generated by various methods. The realism score has been multiplied by 100. ↓/↑ means a smaller/larger value of the metric represents a better performance.

| Method | | Realism | | | | | | | | | | Diversity | | |
|---|---|---|---|---|---|---|---|---|---|---|---|---|---|---|
| | | Agent | | | | | Scene | | | | | Agent | | | Scene |
| | | LO↓ | LA↓ | JE↓ | YR↓ | AVG↓ | LO↓ | LA↓ | JE↓ | YR↓ | AVG↓ | MD↑ | AD↑ | FD↑ | WD↑ |
| Image | SNet | 16.83 | 11.35 | 10.51 | 12.81 | 12.88 | 9.48 | 6.65 | 5.97 | 7.96 | 7.52 | 0.00 | 0.00 | 0.00 | 0.00 |
| | TSim | 20.61 | 18.47 | 8.37 | **1.32** | 12.19 | 10.35 | 9.02 | 12.41 | 7.32 | 9.78 | 0.18 | 0.11 | 0.22 | 1.06 |
| Rule | BITS | 13.75 | 8.68 | 9.53 | 14.65 | 11.65 | 9.42 | 6.11 | 3.44 | 10.42 | 7.35 | 5.32 | 3.83 | 8.24 | 3.95 |
| | CTG | 8.27 | 7.97 | 15.21 | 8.22 | 9.92 | 11.13 | 8.81 | 9.19 | 5.39 | 8.63 | 6.72 | 4.07 | **11.82** | 10.24 |
| Text | LGen | 7.35 | 7.71 | 15.46 | 9.70 | 10.06 | 11.42 | 7.41 | 9.47 | 6.15 | 8.61 | 7.08 | 6.48 | 8.56 | 8.92 |
| | CTG++ | 5.99 | 9.68 | 27.84 | 7.33 | 12.71 | 12.54 | 5.24 | 11.18 | 5.37 | 8.58 | 6.14 | 5.94 | 4.35 | 7.55 |
| | Traj-LLM | **3.72** | **3.82** | **1.54** | 3.85 | **3.24** | **5.65** | **2.42** | **3.21** | **1.56** | **3.21** | **9.33** | **9.62** | 10.89 | **15.92** |

# E    LIMITATION ANALYSIS

In Table 9, we report the proportion of different failure types within failure cases of Traj-LLM. For instance, when Traj-LLM fails to generate an Overtake interaction in the Straightway map, we classify the failure into three typical types: collision (**Collision**), off-road (**Off-road**), and the miss of the specific interaction (**No-Interaction**). We then calculate the proportion of each type for different combinations of Interaction and Map. Figure 14 shows failure cases.

Table 9: Proportion of different types within failure cases of Traj-LLM.

| Interaction | | Straightway | Bend | Roundabout | Cross | Y-shaped | T-shaped |
|---|---|---|---|---|---|---|---|
| **Overtake** | **Collision** | 0.795 | 0.516 | 0.719 | 0.311 | 0.688 | 0.660 |
| | **Off-Road** | 0.045 | 0.042 | 0.094 | 0.070 | 0.020 | 0.181 |
| | **No-Interaction** | 0.160 | 0.442 | 0.187 | 0.619 | 0.292 | 0.159 |
| **Bypass** | **Collision** | 0.693 | 0.642 | 0.300 | 0.581 | 0.685 | 0.961 |
| | **Off-Road** | 0.132 | 0.074 | 0.200 | 0.245 | 0.201 | 0.029 |
| | **No-Interaction** | 0.175 | 0.284 | 0.500 | 0.174 | 0.114 | 0.010 |
| **Yield** | **Collision** | 0.695 | 0.369 | 0.222 | 0.656 | 0.752 | 0.789 |
| | **Off-Road** | 0.024 | 0.331 | 0.334 | 0.315 | 0.067 | 0.132 |
| | **No-Interaction** | 0.281 | 0.300 | 0.444 | 0.029 | 0.181 | 0.079 |

The failure rate of **No-Interaction** is relatively lower than the other two types, highlighting the controllability of Traj-LLM. **Collision** and **Off-Road** occur more frequently.

We conjecture that the above problem mainly stems from the polyline encoder employed by Traj-LLM's failure to interpret complex maps accurately. In our implementation, Traj-LLM heavily relies on a standalone polyline encoder to embed the map information into the hidden feature, which is concatenated with the interaction and behavior features for generating the vehicle trajectories. It means that Traj-LLM does not comprehensively parse the map information. This differs from the text of vehicle interactions and behaviors that are well parsed by Traj-LLM and embedded into features. In the future, we will follow two directions to solve the above problem:

- We can investigate how to transform the map's polylines into text properly. Traj-LLM, which has been pre-trained on a large amount of textual data, may be better able to parse the textual map.
- We can also study how to make Traj-LLM a large visual-language model. The model can regard the map as an image representing not only the lane lines but also other environmental factors like trees, buildings, etc. The model trained on the map images can parse the maps with complex environmental factors, generating the vehicles that fit the maps.

# F    ADDITIONAL VISUAL CASES

In Figures 15 and 16, we visualize the generated trajectories with the interaction of yielding and by-passing. We provide more visualization results of trajectories generated by Traj-LLM in Figure 17. The Traj-LLM method can generate interactions between vehicles and pedestrians, bicycles, and traffic cones (see Figure 18).

In Figure 19, we show the positive influence of adding the trajectories generated by Traj-LLM for training the trajectory prediction model, MTR. In each scenario, we focus on visualizations of each agent's ground-truth trajectory in gray and the one with the highest confidence value in color predicted by MTR. By adding the generated trajectories to the training data, MTR reduces the cases of collision and off-road while predicting the interactive trajectories better than the model without training on the generated data.

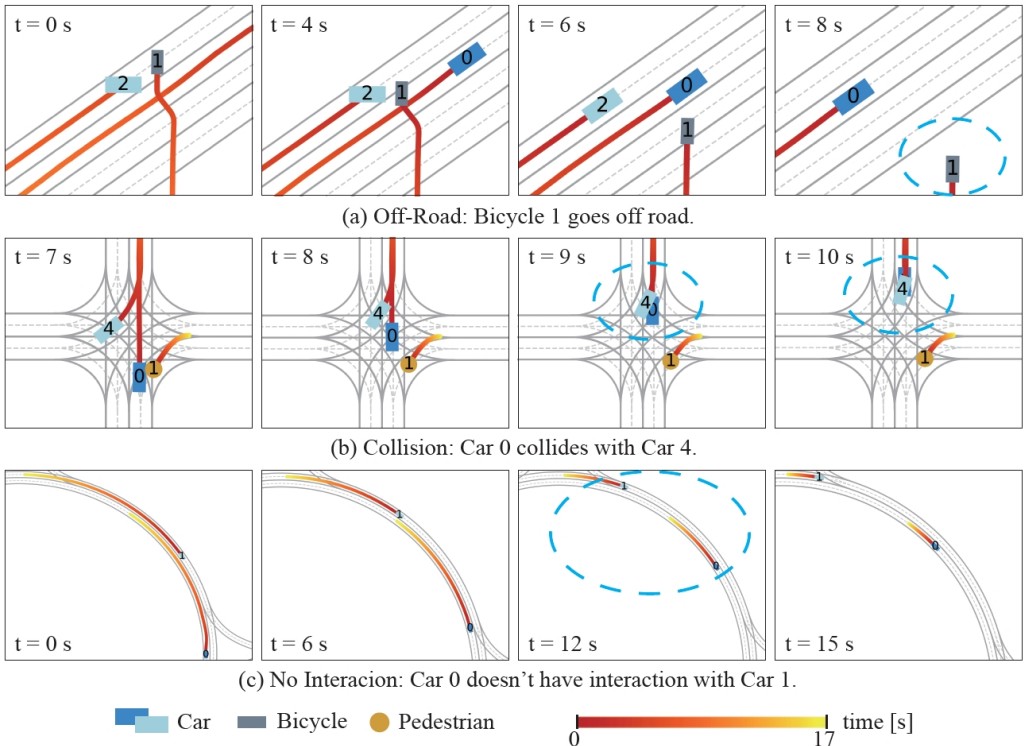

(a) Off-Road: Bicycle 1 goes off road.

(b) Collision: Car 0 collides with Car 4.

(c) No Interacion: Car 0 doesn't have interaction with Car 1.

Car    Bicycle    Pedestrian    time [s]

Figure 14: The visualization of trajectories generated by Traj-LLM that resulted from different failure reasons.

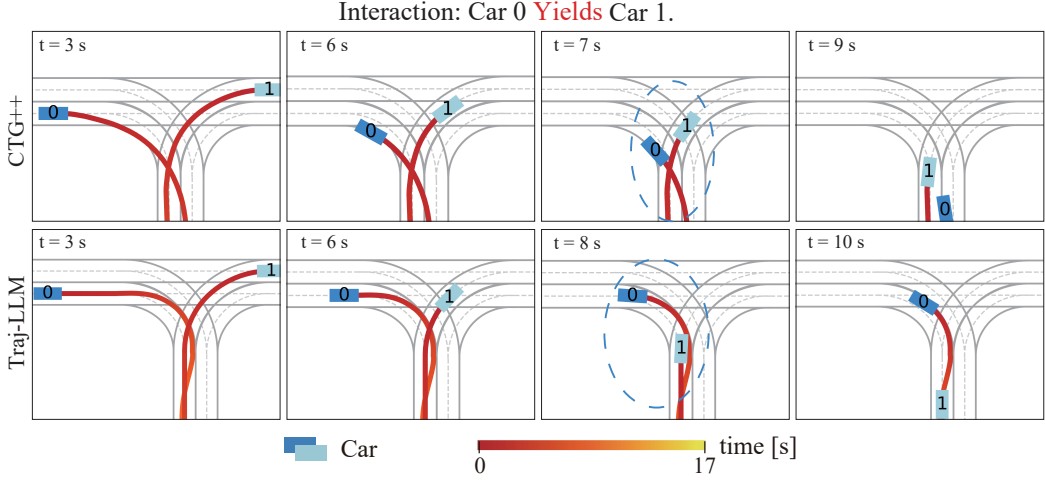

Interaction: Car 0 Yields Car 1.

Car    time [s]

Figure 15: Controllability of CTG++ and Traj-LLM. In this example, Traj-LLM successfully control Car 0 to yield Car 1, while the state-of-the-art CTG++ fails in this case.

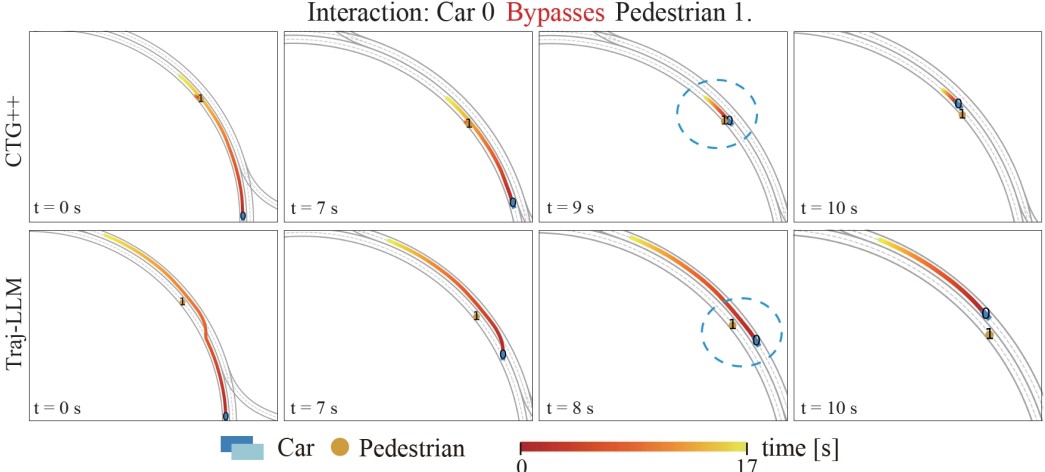

Figure 16: Controllability of CTG++ and Traj-LLM. In this example, Traj-LLM successfully control Car 0 to bypass Car 1, while the state-of-the-art CTG++ fails in this case.

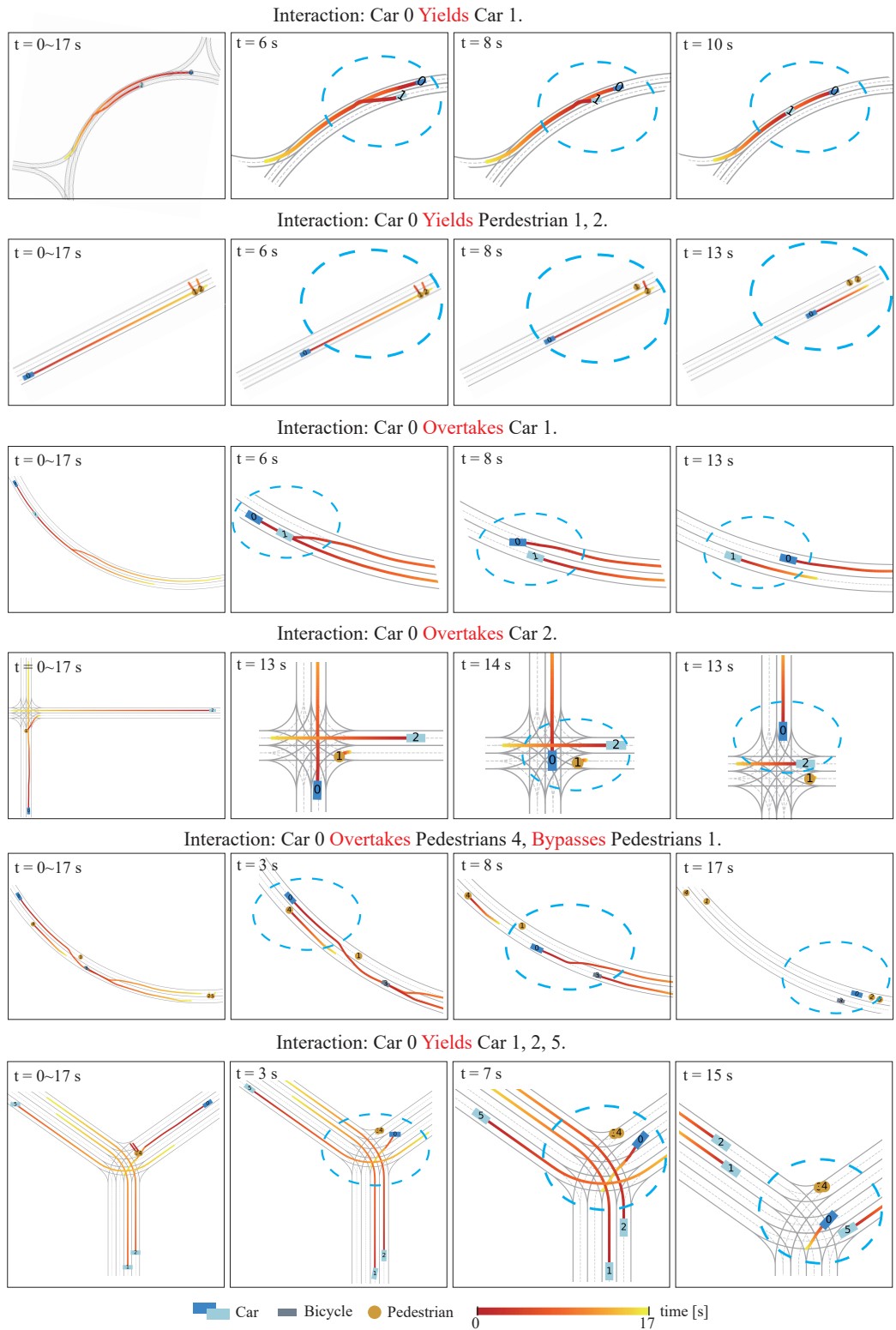

Figure 17: Visualization of trajectories generated by Traj-LLM in various road topologies with different interactions.

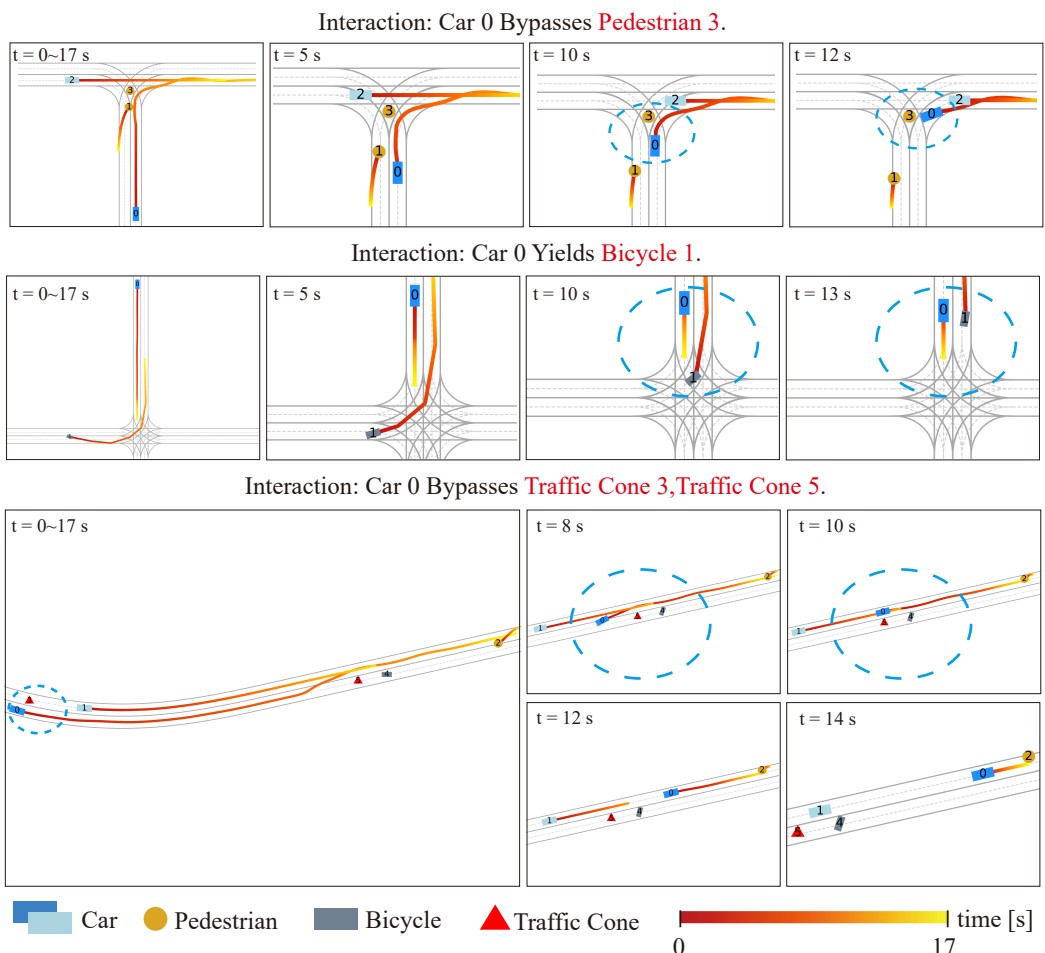

Figure 18: Visualization of trajectories generated by Traj-LLM, which demonstrates its ability to generalize well to scenarios involving traffic cones, bicycles, and pedestrians.

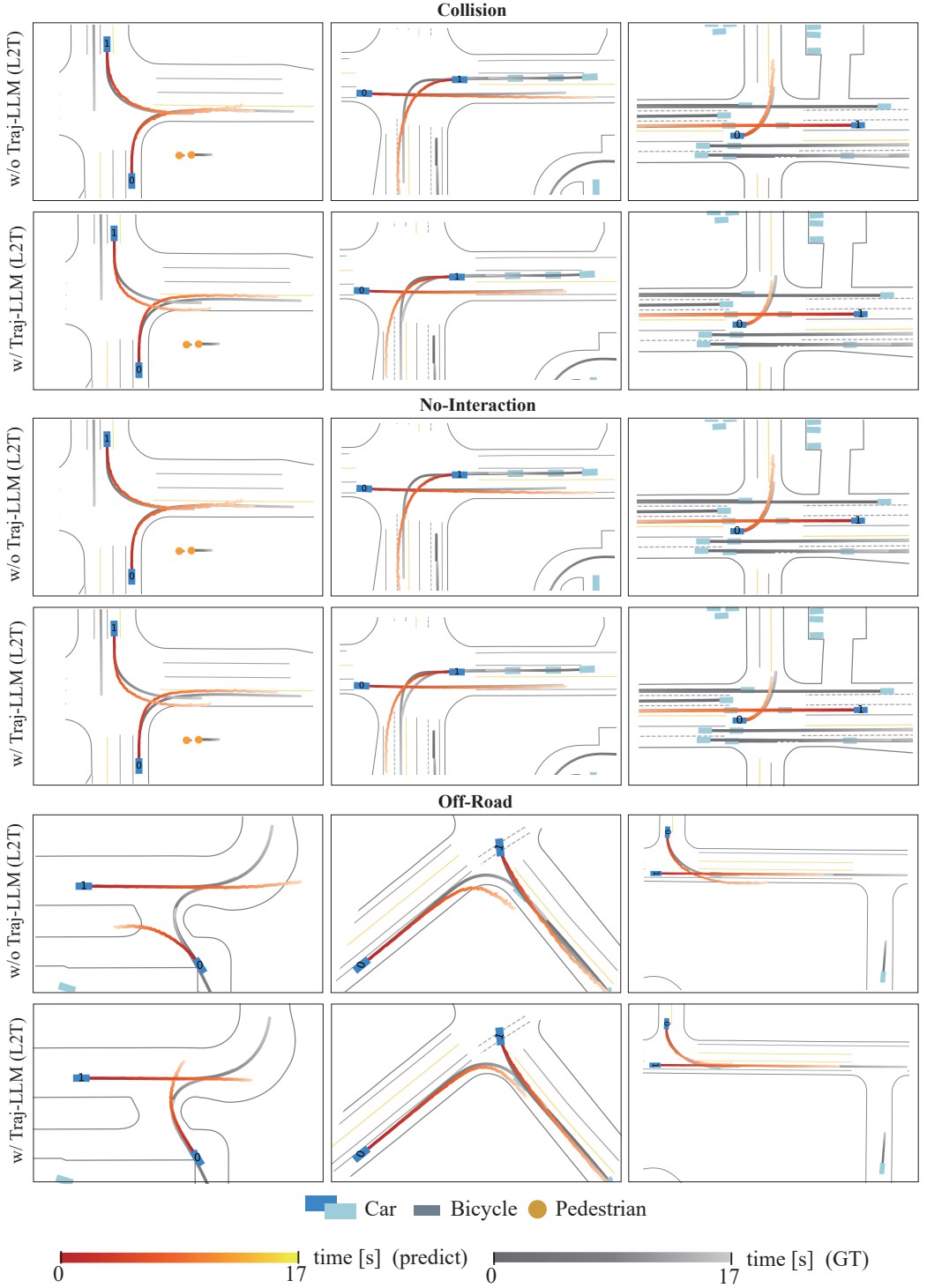

Figure 19: We add trajectories generated by Traj-LLM to train MTR, which reduces **Collision**, **No-Interaction**, and **Off-Road** in the trajectory prediction task.