# OpenReview forum: "Trajectory-LLM: A Language-based Data Generator for Trajectory Prediction in Autonomous Driving"
_ICLR.cc/2025/Conference — ICLR 2025 Poster_

### Official Review · Reviewer_qEvw · 2024-10-31

**Soundness:** 2
**Presentation:** 2
**Contribution:** 2
**Rating:** 6
**Confidence:** 4

**Summary:**

This paper presents a method for generating vehicle trajectories in various traffic scenarios using large language models, employing a hierarchical generation strategy of 'interaction-behavior-trajectory.' Additionally, this work creates the L2T dataset to train the proposed model. Experimental results show that the motion prediction model trained with the generated data performs better on the mAP metric compared to the model trained solely on WOMD, providing preliminary validation of the method's effectiveness.

**Strengths:**

1. The paper is well-organized and easy to read.
2. The proposed method may help alleviate the reliance of existing motion prediction models on large amounts of real data, facilitating the development of these models.

**Weaknesses:**

1. I did not see which specific large language model was used; the only information provided in the system diagram shows the LLM has an encoder-decoder structure, which does not seem to align with GPT-like models. I urge the authors to elaborate on the design of the large language model.
2. As a generative model designed for trajectory prediction tasks, the experimental section is insufficiently thorough. Firstly, it only selects the WOMD dataset and does not test on other representative datasets, such as Argoverse. Secondly, it only uses mAP as the metric, neglecting other common metrics such as distance error and miss rate. Therefore, the generalizability of the proposed method remains questionable.
3. Training a generative model on a third-party dataset and then using the generated data for model training—does this yield effects equivalent to training on the original third-party dataset? I recommend the authors include an ablation study comparing the experimental results of training with the original dataset.

**Questions:**

1. Can the proposed model be used for simulating traffic scene? I suggest the authors consider this application area, rather than limiting the focus solely to trajectory prediction tasks.

---

> ### Author Response · Authors · 2024-11-22
> **Thanks for your valuable comments!**
>
> We sincerely thank you for your valuable comments. Below, we respond to your questions point-to-point.
>
> **Q1:** I did not see which specific large language model was used; the only information provided in the system diagram shows the LLM has an encoder-decoder structure, which does not seem to align with GPT-like models. I urge the authors to elaborate on the design of the large language model.
>
> **Answer:**  We respectfully clarify that the specific LLM used in this paper is Llama-7B (see Section C of the supplementary file). Furthermore, we have provided an anonymous GitHub repository (https://github.com/anonymous-github-Traj-LLM/Traj-LLM) for the complete implementation of Traj-LLM to help readers to follow our work.
>
> We sincerely thank you for pointing out this confusion about our framework's encoder-decoder structure. Based on the fundamental Llama-7B, Traj-LLM follows the decoder-only structure like GPT. In Figure 2 of our framework, the encoder embeds the texts of vehicle interactions and behaviors into hidden features. It actually acts as the text embedding layer. To eliminate the confusion, we have modified the encoder in Figure 2 and its definition in the relevant formulations to the text embedding layer.
>
>
> **Q2:** As a generative model designed for trajectory prediction tasks, the experimental section is insufficiently thorough. Firstly, it only selects the WOMD dataset and does not test on other representative datasets, such as Argoverse. Secondly, it only uses mAP as the metric, neglecting other common metrics such as distance error and miss rate. Therefore, the generalizability of the proposed method remains questionable.
>
> **Answer:**  Thanks for your suggestion. We have added the metrics of distance error and miss rate to the results on the Waymo dataset (see Table 4 of the paper and Tables 5-7 of the revised supplementary file). We have also added the trajectory prediction performances on the Argoverse dataset, where the vehicle trajectories generated by Traj-LLM also improve the downstream prediction model (see Table 7 of the revised supplementary file).
>
> **Q3:** Training a generative model on a third-party dataset and then using the generated data for model training—does this yield effects equivalent to training on the original third-party dataset? I recommend the authors include an ablation study comparing the experimental results of training with the original dataset.
>
> **Answer:**  We respectfully note that the experimental results in Table 5 of the revised supplemental file compare the training on the original and generated data. As shown in the first row (the original L2T dataset) and the fourth row (the data generated by Traj-LLM), training on the generated data yields better results. For a fair comparison, the amounts of original and generated samples for training are the same. This performance improvement stems from the data diversity brought by the generation process of Traj-LLM, which helps the downstream prediction model to witness vehicle trajectories with novel shapes. We remark that the data diversity of our generated trajectories has been evidenced in Table 1 of the paper.
>
> **Q4:**  Can the proposed model be used for simulating traffic scene? I suggest the authors consider this application area, rather than limiting the focus solely to trajectory prediction tasks.
>
> **Answer:**  Thanks for sharing this inspiring idea. Without needing to modify the main model architecture, a trivial way of extending Traj-LLM to simulate the traffic scene is to input the interactions between every pair of vehicles to generate all vehicles' trajectories. In this paper, we have experimented with using Traj-LLM to yield the trajectories of up to 5 interactive vehicles in the same scenario, which is already the basic traffic scene.
>
>
> The above trivial solution may need complex text descriptions of vehicle interactions, especially for traffic scenes containing many vehicles. A possible solution is to prepare higher-level descriptions of traffic scenes in short texts for training Traj-LLM, rather than using the complex text descriptions of all vehicle interactions. This could be investigated in our future work. It also remains a problem of efficiently combining the scene- and vehicle-level descriptions in a short text, which Traj-LLM parses to control vehicle interactions to make them consistent in the same scene to avoid collision or out-of-map.

---

> > ### Comment · Reviewer_qEvw · 2024-11-24
> > **Thanks for your clear clarification.**
> >
> > Thanks for your clear clarification and I will raise my score.

---

> > > ### Author Response · Authors · 2024-11-25
> > >
> > > Dear Reviewer qEvw,
> > >
> > > Thank you again for your review. We are pleased to see that your questions have been answered.
> > >
> > > Best,
> > >
> > > Authors of Paper ID 4151

---

### Official Review · Reviewer_v7Vc · 2024-11-01

**Soundness:** 3
**Presentation:** 3
**Contribution:** 2
**Rating:** 6
**Confidence:** 4

**Summary:**

This paper introduces Trajectory-LLM (Traj-LLM), which leverages Large Language Models to generate realistic vehicle trajectories from textual descriptions. The method employs an "interaction-behavior-trajectory" process to translate vehicle interactions into trajectories, ensuring the realism and controllability of the trajectories.

**Strengths:**

* Well-written paper. Easy to follow.
* The proposed "interaction-behavior-trajectory" process to guide the trajectory generation can help ensure better realism and controllability.
* The author contributed a new dataset L2T.

**Weaknesses:**

* The proposed dataset L2T is collected in simulation platform and manually use the wheels to drive and collect the data. The realism of the proposed dataset is questionable. Further justification and motivations are needed for creating a new dataset with good realism instead of curating existing open real-world dataset to create such paired dataset.
* The main benchmark is performed on the proposed L2T and open dataset WOMD is only used in ablation study while  existing literature mainly compared on other open datasets. Without benchmarking on the same open dataset, it would be hard to justify the technical contribution of the proposed model.

**Questions:**

Please see above sections.

---

> ### Author Response · Authors · 2024-11-23
> **Thanks for your valuable comments!**
>
> We sincerely thank you for your valuable comments. Below, we respond to your questions point-to-point.
>
> **Q1:** The proposed dataset L2T is collected in simulation platform and manually use the wheels to drive and collect the data. The realism of the proposed dataset is questionable. Further justification and motivations are needed for creating a new dataset with good realism instead of curating existing open real-world dataset to create such paired dataset.
>
> **Answer:** We sincerely appreciate your feedback on our dataset. We respectfully clarify that many vehicle trajectories in datasets like Waymo and Nuscenes lack or only have weak interaction. This phenomenon is mainly attributed to the high risk of traffic accidents associated with collecting vehicle trajectories with high-intensity interactions on real roads. On the other hand, vehicle trajectories with high-intensity interactions play a crucial role in training autonomous driving systems, as they help the systems to reasonably handle many vehicle interactions that are prone to causing traffic accidents.
>
> Rather than taking the risk of collecting data on the real-world road, we resort to simulating the vehicle trajectories with high-intensity interactions in the virtual world. We construct a new simulator to increase the realism of vehicle trajectories collected in the simulated environment. Our simulator integrates the re-developed hardware and software to efficiently collect and annotate the vehicle trajectories in the L2T dataset. We have added more details about our simulator to Section A.1 of the revised supplementary file.
>
>
> The hardware component of our simulator mainly includes a steering wheel, a gear shifter, gas and brake pedals, and several displays, as shown in Figure 7(a) of the supplementary file. The displays visualize the first-person view of the road environments from the left, right, front, and back car sides, as shown in Figure 7(b). Here, the road environments are simulated by LGSVL, a simulation engine for autonomous driving. Our hardware simulates a realistic car cockpit, helping users to drive on the virtual road to produce vehicle trajectories as real as possible.
>
> The software component is mainly built on top of the LGSVL engine. LGSVL can record all trajectories of vehicles traveling on the virtual road. We re-develop this engine to enable the connection among multiple sets of the above hardware, thus allowing multiple users to control different cars on the same virtual road. This software facilitates the simulation of vehicle interactions like those in the real world. In Figure 8, we show the interactive cars in the virtual road from the third-person view.
>
> In Section A.3 of the revised supplementary file, we have compared the data distribution of our L2T dataset and other real-world datasets like Waymo. In Figure 12, we compare the distributions of trajectories' shapes represented by the waypoints' coordinates. The distributions are similar, showing that the vehicle trajectories in the L2T dataset are realistic.
>
> **Q2:** The main benchmark is performed on the proposed L2T and open dataset WOMD is only used in ablation study while existing literature mainly compared on other open datasets. Without benchmarking on the same open dataset, it would be hard to justify the technical contribution of the proposed model.
>
>
> **Answer:** Thanks for your constructive comment. As you requested, we have compared different trajectory generation models on the real-world Waymo dataset to measure the realism and diversity of the generated trajectories. The experimental results are in Table 8 of the revised supplementary file, where Traj-LLM outperforms other methods. These results show that Traj-LLM's generated trajectories also deliver realism and diversity compared to real-world data rather than the simulated L2T dataset only.
>
> Apart from the downstream trajectory prediction on the Waymo dataset in Table 4 of the revised paper, we have added the trajectory prediction performances on the Argoverse dataset, where the vehicle trajectories generated by Traj-LLM also improve the state-of-the-art prediction model (see Table 7 of the revised supplementary file). These results show that the vehicle trajectories generated by Traj-LLM deliver generalization across different datasets like Waymo and Argoverse.

---

> ### Author Response · Authors · 2024-11-26
> **Sincerely Request Your New Comments**
>
> Dear Reviewer v7Vc,
>
> We thank you again for your valuable comments, which significantly helped us polish our paper. Could you kindly let us know if the responses have addressed your concerns and if further explanations or clarifications are needed? Your time and efforts in evaluating our work are appreciated greatly.
>
> Best,
>
> Authors of Paper ID 4151

---

> ### Author Response · Authors · 2024-12-03
> **Thanks for your feedback!**
>
> Dear Reviewer v7Vc,
>
> Thank you again for your review. We are pleased to see that your questions have been answered. We will follow your suggestion to add more detailed information to our paper.
>
> Best,
>
> Authors of Paper ID 4151

---

### Official Review · Reviewer_9AVa · 2024-11-04

**Soundness:** 3
**Presentation:** 3
**Contribution:** 2
**Rating:** 6
**Confidence:** 5

**Summary:**

This paper adopts large language models (LLMs) to generate the trajectories for autonomous driving. In this regard, they explore the interaction-behavior-trajectory translation process in contrast to only interaction-trajectory, which is widely used for other LLMs-based approaches. Their method comprises two stages, where in the first stage, the textual descriptions of vehicle interaction are translated to behaviors, and in the second stage, those behaviors are converted into specific motion parameters for trajectories. Based on this, the authors have also created a new Language-to-Trajectory (L2T) dataset. The proposed method has shown promising results on L2T and the Waymo Motion Prediction datasets compared to state-of-the-art methods.

**Strengths:**

Here are the strengths of this paper:
1. The inclusion of driving behaviors for trajectory prediction is promising.
2. Another promising aspect of this paper is the creation of a new benchmark dataset, L2T.
3. Overall, the presentation quality of the paper is good. However, some grammatical mistakes can be corrected afterwards.

**Weaknesses:**

However, the paper has shown better performances in the evaluation, yet some comments need to be addressed:
1. I acknowledge that the authors have put great effort into creating the dataset, but it was unclear whether they have used existing datasets, for instance, Waymo and Nuscenes, to build their dataset. If this is not the case, and they adopt some manual feedback from the driver, how can they evaluate those behavior's annotations?
2. The authors have experimented with the proposed method in open-loop settings; it would be good to see how the proposed method performs in the closed-loop evaluation.
3. It is good to see that authors have also dedicated a section in supplementary material for the limitation of their work, but it would be good to see what the future directions will be based on their work.
4. In the paper, the author has used random locality attention; it would be good to do an ablation study on its effect on driving behavior and also the trajectories. Also, if another form of attention, for instance, cross attention, is used, what will affect behavior and trajectories?

**Questions:**

However, the paper has shown better performances in the evaluation, yet some comments need to be addressed:
1. I acknowledge that the authors have put great effort into creating the dataset, but it was unclear whether they have used existing datasets, for instance, Waymo and Nuscenes, to build their dataset. If this is not the case, and they adopt some manual feedback from the driver, how can they evaluate those behavior's annotations?
2. The authors have experimented with the proposed method in open-loop settings; it would be good to see how the proposed method performs in the closed-loop evaluation.
3. It is good to see that authors have also dedicated a section in supplementary material for the limitation of their work, but it would be good to see what the future directions will be based on their work.
4. In the paper, the author has used random locality attention; it would be good to do an ablation study on its effect on driving behavior and also the trajectories. Also, if another form of attention, for instance, cross attention, is used, what will affect behavior and trajectories?

---

> ### Author Response · Authors · 2024-11-20
> **(1/2) Thanks for your valuable comments!**
>
> We sincerely thank you for your valuable comments. Below, we respond to your questions point-to-point.
>
> **Q1:** I acknowledge that the authors have put great effort into creating the dataset, but it was unclear whether they have used existing datasets, for instance, Waymo and Nuscenes, to build their dataset.
>
> **Answer:** We sincerely appreciate your feedback on our dataset. We respectfully clarify that many vehicle trajectories in datasets like Waymo and NuScenes lack or only have weak interaction. This phenomenon is mainly attributed to the high risk of traffic accidents associated with collecting vehicle trajectories with high-intensity interactions on real roads. On the other hand, vehicle trajectories with high-intensity interactions play a crucial role in training autonomous driving systems, as they help the systems to reasonably handle many vehicle interactions that are prone to causing traffic accidents. Thus, we do not use these existing datasets to build our dataset, where our dataset focuses on high-intensity vehicle interactions.
>
> Rather than taking the risk of collecting data on the real-world road like the existing datasets, we resort to simulating the vehicle trajectories with high-intensity interactions in the virtual world. We construct a new simulator to increase the realism of vehicle trajectories collected in the simulated environment. Our simulator integrates the re-developed hardware and software to efficiently collect and annotate the vehicle trajectories in the L2T dataset. We have added more details about our simulator to Section A.1 of the revised supplementary file.
>
> The hardware component of our simulator mainly includes a steering wheel, a gear shifter, gas and brake pedals, and several displays, as shown in Figure 7(a) of the supplementary file. The displays visualize the first-person view of the road environments from the left, right, front, and back car sides, as shown in Figure 7(b). Here, the road environments are simulated by LGSVL, a simulation engine for autonomous driving. Our hardware simulates a realistic car cockpit, helping users to drive on the virtual road to produce vehicle trajectories as real as possible.
>
> The software component is mainly built on top of the LGSVL engine. LGSVL can record all trajectories of vehicles traveling on the virtual road. We re-develop this engine to enable the connection among multiple sets of the above hardware, thus allowing multiple users to control different cars on the same virtual road. This software facilitates the simulation of vehicle interactions like those in the real world. In Figure 8, we show the interactive cars in the virtual road from the third-person view.
>
> In Section A.3 of the revised supplementary file, we have compared the data distribution of our L2T dataset with other real-world datasets like Argoverse and Waymo. In Figure 12, we compare the distributions of trajectories' shapes represented by the waypoints' coordinates. The distributions are similar, showing that the vehicle trajectories in the L2T dataset are realistic.
>
> **Q2:** If this is not the case, and they adopt some manual feedback from the driver, how can they evaluate those behavior's annotations?
>
> **Answer:** Section A.1 of the revised supplementary file introduces how to evaluate the annotation quality.  We recruit 30 drivers with rich experience to explain and document vehicle interactions and behaviors with driving logic. We conduct a two-level check to evaluate the annotation quality. We develop an automated Python script at the first check level to verify whether the trajectories align with the text interactions and behaviors. These scripts incorporate some basic checking rules, such as the rule that the overtaking between two vehicles must involve acceleration, and the behavior parameters within the acceleration must be equivalent to the motion parameters of the trajectories. We have uploaded these scripts to the anonymous GitHub repository (https://github.com/anonymous-github-Traj-LLM/Traj-LLM/tree/master/auto_behavior_detection).
>
> At the second check level,  we divided the scenarios that have passed the first check level into 10 groups. Three drivers check the annotations of each group of scenarios. We randomly select 20% of scenarios from each group. If half of the selected scenarios have false annotations, all scenarios in this group must be re-annotated; otherwise, only the scenarios with false annotations are re-annotated.

---

> ### Author Response · Authors · 2024-11-20
> **(2/2) Thanks for your valuable comments!**
>
> **Q3:** The authors have experimented with the proposed method in open-loop settings; it would be good to see how the proposed method performs in the closed-loop evaluation.
>
> **Answer:** Thank you for sharing this inspiring idea. Here, we respectfully clarify the central purpose of this paper, which is to find a cost-effective way of collecting a large amount of vehicle trajectories with high-intensity interactions. Rather than collecting these trajectories by taking the risk of traffic accidents, we employ LLM, the most advanced language-understanding technique, to parse the brief text of vehicle interactions to generate the trajectories. These generated trajectories play as the data for training the downstream autonomous driving models like trajectory prediction. From the view of the application of generated trajectory data for the downstream model training, LLM generates trajectory data in the open- or closed-loop does not matter. This is because the data generation and downstream model training are separate, which means the downstream model training is unaware of the open- or closed-loop generation. Thus, this paper focuses more on increasing the realism, diversity, and controllability of the trajectory generation process, which are more relevant for the performance improvement on the downstream model.
>
> We agree with the reviewer's view of making LLM work in the closed-loop setting. This setting may allow LLM to generate vehicle trajectories that represent interactions under the circumstances with unpredictable changes, thus helping to train more robust downstream models. However, letting LLM work in the closed-loop setting means LLM has to interrupt its generation based on the current environment as input and start a new reasoning using the changed environment. This process results in problems such as the waste of computations performed before the interruption and the slow speed of data generation. In contrast to applying LLM to understand human dialog, the above problems relevant to autonomous driving become particularly prominent when the road environment is complex and ever-changing. They must be solved perfectly before implementing the closed-loop LLM to generate trajectory data.
>
> **Q4:** It is good to see that authors have also dedicated a section in supplementary material for the limitation of their work, but it would be good to see what the future directions will be based on their work.
>
> **Answer:** Thank you for pointing out this loss of information. We have modified our discussion of the method's limitation in Section E of the supplementary file. Traj-LLM relies heavily on a standalone polyline encoder to embed the map information into the hidden feature, which is concatenated with the interaction and behavior features for generating the vehicle trajectories. It means that Traj-LLM does not comprehensively parse the map information. This differs from the text of vehicle interactions and behaviors that are well parsed by Traj-LLM and embedded into features. In the future, we will follow two directions to solve the above problem:
> - We can investigate how to transform the map's polylines into text properly. Traj-LLM, which has been pre-trained on a large amount of textual data, may be better able to parse the textual map.
> - We can also study how to make Traj-LLM a large visual-language model. In this manner, the model can regard the map as an image representing not only the lane lines but also other environmental factors like trees, buildings, etc. The model trained on the map images can parse the maps with complex environmental factors, generating the vehicles that fit the maps.
>
> **Q5:** In the paper, the author has used random locality attention; it would be good to do an ablation study on its effect on driving behavior and also the trajectories. Also, if another form of attention, for instance, cross attention, is used, what will affect behavior and trajectories?
>
> **Answer:** Thanks for your valuable advice. In the third row of Table 3 in Section 5.2, we remove the random and locality properties to degrade the random locality attention to a vanilla cross-attention, yielding lower realism, diversity, and controllability than the random locality attention.  We have modified the text of Section 5.2 to clarify the above point. We have also conducted the ablation study by removing random or locality, reporting the results from the second to the sixth rows of Table 3. These alternatives degrade the quality of trajectory generation. The experimental results demonstrate the effectiveness of enabling the random and locality together.

---

> ### Comment · Reviewer_9AVa · 2024-11-26
>
> Thank you for providing the detailed response. Well the authors have mentioned in their response as "We respectfully clarify that many vehicle trajectories in datasets like Waymo and NuScenes lack or only have weak interaction. This phenomenon is mainly attributed to the high risk of traffic accidents associated with collecting vehicle trajectories with high-intensity interactions on real roads."
> However, these statements appear contradictory. Waymo and NuScenes are widely adopted datasets, and extensive research has utilized them for motion planning and pedestrian interaction. In the subsequent paragraph, the authors mention using a simulator to collect their dataset. While it is true that simulators provide inherent safety advantages, given the availability of well-established interaction datasets, the question arises: what is the specific impact or added value of this dataset?
> One more concern that is already raised by one of the reviewer is the evaluation of proposed method on other open-loop dataset for instance NuScenes. Also one minor or on a presentation side comment is that authors should put the necessary (that helps reader to understand) information in the main paper rather than in supplementary material. I think supplementary material is just to complement your paper.

---

> ### Author Response · Authors · 2024-11-28
> **Sincerely thank you for your valuable feedback!**
>
> **Q1:**  Well the authors have mentioned in their response as "We respectfully clarify that many vehicle trajectories in datasets like Waymo and NuScenes lack or only have weak interaction. This phenomenon is mainly attributed to the high risk of traffic accidents associated with collecting vehicle trajectories with high-intensity interactions on real roads." However, these statements appear contradictory. Waymo and NuScenes are widely adopted datasets, and extensive research has utilized them for motion planning and pedestrian interaction.
>
> **Answer:** We are sorry for the misunderstanding raised by our previous statement. In Section 4 of our paper, we explained the interactive characteristics of the L2T dataset, describing them as "These high-intensity interactions should be completed in a short range of less than 10 meters." In contrast, within real-world datasets such as Waymo and NuScenes, most interactions are maintained at a safe distance (>15m), which we call weak interactions. A detailed explanation is in the next answer to Q2.
>
>
> **Q2:**  In the subsequent paragraph, the authors mention using a simulator to collect their dataset. While it is true that simulators provide inherent safety advantages, given the availability of well-established interaction datasets, the question arises: what is the specific impact or added value of this dataset?
>
> **Answer:** As previously mentioned in the Answer of Q1, the L2T dataset is characterized by short-range, high-intensity interactions. In Figure 13 of the supplementary material, we have visualized this discrepancy. It shows that the shortest interaction distances in L2T are more concentrated within the 0-10 meter range, whereas WOMD concentrates within the 15-25 meter range. It demonstrates that interactions in L2T are stronger than WOMD. The characteristic of longer interaction distances in WOMD primarily stems from cautious and risk-averse drivers in the real world. Autonomous driving systems often require a greater capacity to handle high-intensity interactions, which is one of the significances of our dataset. In Table 4-7 of our paper, we demonstrated that adding L2T or generated data from Traj-LLM into the training of motion prediction models can improve their performance on WOMD and Argoverse.
>
> On the other hand, L2T also has rich annotations of traffic scenarios. Different from existing datasets, L2T not only contains high-level annotations about interaction relationships (bypass, overtake, yield) and driving targets (go straight, turn left, turn right), but also includes detailed driving behavior annotations such as cruise and change lane. In the framework of Traj-LLM, we conduct the“interaction-behavior-trajectory" translation and achieve better performance in Realism, Diversity, and Controllability. We hope that L2T can contribute to exploring the enhancement of performance and interpretability of adding textual information in autonomous driving systems. It is also one of our future directions.

---

> > ### Author Response · Authors · 2024-12-01
> > **Sincerely thank you for your valuable feedback!**
> >
> > **Q3:**  One more concern that is already raised by one of the reviewer is the evaluation of proposed method on other open-loop dataset for instance NuScenes.
> >
> > **Answer:** In response to the concerns raised by reviewer v7Vc, we have compared different trajectory generation models on the real-world Waymo dataset. The results are in Table 8 of the revised supplementary file, where Traj-LLM outperforms other methods. Here we present the results of comparing different trajectory generation models on the NuScenes dataset. In summary, we have conducted evaluations on the simulated L2T dataset (see Table 1 in the main paper), the real-world WOMD (see Table 8 in the revised supplementary materials), and the real-world NuScenes (see Table Q3 below). These results indicate that Traj-LLM outperforms other methods in both Realism and Diversity.
> >
> > ### Table Q3
> > | Method    | Agent Realism| | | | | Scene Realism | | | | | Agent Diversity | | | Scene Diversity |
> > |-|-|-|-|-|-|-|-|-|-|-|-|-|-|-|
> > |           | LO    | LA    | JE    | YR    | AVG       | LO    | LA    | JE    | YR    | AVG       |   MD      | AD    | FD    | WD    |
> > | SNet      | 10.10 | 8.83 | 7.33 | 7.73 | 8.50 | 9.60 | 8.99 | 7.13 | 8.50 | 8.56 | 0 | 0 | 0 | 0 |
> > | TSim      | 19.53 | 16.25 | 9.98 | 2.70 | 12.12 | 27.67 | 27.21 | 11.46 | 1.85 | 17.05 | 3.16 | 1.06 | 2.49 | 3.07 |
> > | BITS      | 8.54 | 11.04 | 8.67 | 8.33 | 9.15 | **7.28** | 7.02 | 8.05 | 8.48 | 7.71 | 4.77 | 2.67 | 6.61 | 7.76 |
> > | CTG       | 14.44 | 12.24 | 14.79 | 3.05 | 11.13 | 14.10 | 11.18 | 13.35 | 3.12 | 10.44 | 8.30 | 4.55 | 8.01 | 9.38 |
> > | LGen      | 15.04 | 12.62 | 17.49 | 3.72 | 12.22 | 11.93 | 16.39 | 15.18 | 5.27 | 12.19 | 9.42 | 6.75 | 2.18 | 9.91 |
> > | CTG++     | 17.30 | 14.47 | 15.51 | 3.53 | 12.70 | 16.25 | 12.36 | 14.08 | 4.26 | 11.74 | 7.10 | 8.90 | 1.51 | 1.03 |
> > | Traj-LLM  | **4.28** | **6.74** | **5.90** | **2.18** | **4.78** | 9.16 | **5.94** | **3.41** | **1.47** | **5.00** | **13.88** | **10.57** | **9.89** | **11.67** |
> >
> >
> >
> > **Q4:**  Also one minor or on a presentation side comment is that authors should put the necessary (that helps reader to understand) information in the main paper rather than in supplementary material. I think supplementary material is just to complement your paper.
> >
> > **Answer:** We sincerely appreciate your suggestions and agree with your opinion. Due to the page limits, we only presented the core results within the main paper. We respectfully invite you to specify which part of supplementary material is necessary to put into the main paper. This will greatly help us to make our paper more understandable.

---

> > ### Comment · Reviewer_9AVa · 2024-12-01
> >
> > Thank you for your response.
> >
> > However, I have some concerns regarding the limitations of this dataset compared to existing ones that include interaction studies. For example, in addition to NuScenes, the INTERACTION dataset is available and comes integrated with a simulator. How does this dataset compare to those alternatives? Another issue, also raised by other reviewers, is the diversity and realism of this dataset. For these reasons, I remain unconvinced and feel that further theoretical and experimental justification is necessary to demonstrate the utility and usability of this dataset. Consequently, based on my initial assessment, I will be revising my rating downward.

---

> > > ### Author Response · Authors · 2024-12-03
> > > **Thanks for your valuable comments!**
> > >
> > > **Q1:**  However, I have some concerns regarding the limitations of this dataset compared to existing ones that include interaction studies. For example, in addition to NuScenes, the INTERACTION dataset is available and comes integrated with a simulator. How does this dataset compare to those alternatives?
> > >
> > > **Answer:** The INTERACTION dataset is a high-quality motion dataset focused on interactive driving scenarios and complex driving behaviors, compared to nuScenes or Argo. However, It is also collected from real-world scenarios.  As in the discussion of our previous response to [Q2](https://openreview.net/forum?id=UapxTvxB3N&noteId=PvOEfA1C1X), L2T is distinguished from other real-world datasets by its short-range, high-intensity interactions that are infrequently encountered in the real world.
> > >
> > > The simulator of the INTERACTION dataset is constructed based on real scenarios from the dataset and is primarily used for closed-loop evaluation, while our simulator integrates the re-developed hardware and software to efficiently collect high-intensity interaction data with real-person driving. In Section A.1 and Figure 7 of the revised supplementary material, we provide a detailed explanation of the collection and annotation of L2T. During our data collection process, the simulator serves as a substitute for real-world environments, with the primary driving actions executed by real drivers. This is because, for the collection of high-intensity interaction data, the simulated environment can completely disregard safety considerations and efficiently filter collision scenarios. The simulator associated with the INTERACTION dataset, due to its limited functionality, cannot serve as a platform for real drivers to execute complex driving behaviors.
> > > We conduct more quantitative comparisons between NuScenes, INTERACTION, and L2T in the table of subsequent Q2 below.

---

> > > > ### Author Response · Authors · 2024-12-03
> > > > **Thanks for your valuable comments!**
> > > >
> > > > **Q2:**  Another issue, also raised by other reviewers, is the diversity and realism of this dataset. For these reasons, I remain unconvinced and feel that further theoretical and experimental justification is necessary to demonstrate the utility and usability of this dataset.
> > > >
> > > >
> > > > **Answer:** Reviewer v7Vc raised the concern about the diversity and realism of L2T,  We have provided a detailed explanation in our response to reviewer v7Vc's [Q1](https://openreview.net/forum?id=UapxTvxB3N&noteId=wzHXZX8B0s) and obtained the approval of reviewer v7Vc, who increased the final rating from 5 to 6.
> > > >
> > > > Another request from reviewer v7Vc was "Providing more quantitative statistics to support the claim of behavioral diversity and realism in the simulated dataset".  We have included additional quantitative statistics and a detailed explanation in our response to [Q3](https://openreview.net/forum?id=UapxTvxB3N&noteId=UFehRk0JXY). These responses have been acknowledged by [reviewer v7Vc](https://openreview.net/forum?id=UapxTvxB3N&noteId=KoJAxtwwbv). For your convenience, we have pasted the response to Q3 from reviewer v7Vc here and additionally included a comparison with the NuScenes and INTERACTION datasets in Table Q3 below. The results in Table Q3 below demonstrate that L2T exhibits closer interaction distances and more frequent interaction behaviors compared to other datasets.
> > > >
> > > >
> > > > > **(Reviewer v7Vc) Q3**: Providing more quantitative statistics to support the claim of behavioral diversity and realism in the simulated dataset would significantly strengthen the argument.
> > > > >
> > > > > **Answer:** In Figure 11 and Figure 12 of the revised supplementary material, we demonstrate the similarity in trajectory shapes, prediction numbers, and trajectory density distributions between L2T and Waymo. In Section A.4, we proposed a novel metric, the closest interaction distance (CID), to denote the closest distances between two vehicles during the entire movement process. The lower the value of CID, the greater the interaction intensity between two vehicles. In Figure 13 of the supplementary material, we visually present the differences in the closest interaction distances between L2T and WOMD. In Table Q3 below, we provide quantitative statistical results for comparing CID and the frequency of different interaction types of WOMD, WOMD-interactive, Argoverse, and L2T.
> > > > >
> > > > > Here we provide the details about the calculation. For each scenario within the dataset, assuming the total number of objects is denoted by $N$ and the number of objects to be predicted is denoted by $a$, the potential number of interactions within that scenario is given by $a \times (N - 1)$. This implies that each predicted object may interact with every other object in the scenario. We then filter based on the CID between two objects, only less than 50 meters are considered to potentially have an interaction. Subsequently, for each pair of objects, the CID can be calculated and the interaction types can be categorized accordingly. All the interaction types are judged by the same automated behavior detection tool. We have uploaded the Python script to the anonymous GitHub repository (https://github.com/anonymous-github-Traj-LLM/Traj-LLM/tree/master/statistics).
> > > > >
> > > > > The results show that L2T exhibits a higher frequency of interactive behaviors and shorter interaction distances. All these results indicate that the L2T dataset possesses behavioral diversity and realism. We sincerely appreciate the suggestion from the reviewer, the generation of complex interactions between more agents is also one of the directions of our future work.
> > > > >
> > > > > On the other hand, regarding the collection of strong-interaction trajectories,  the advantages of simulation are irreplaceable. We can utilize simulation platforms to generate plenty of short-range interactions and conveniently filter out trajectories that collide with each other, without considering safety concerns. This also underscores the significant difficulty in collecting such data in the real world.
> > > >
> > > > Table Q3
> > > >
> > > > | Dataset | minCID | medianCID | meanCID | Bypass Freq | Overtake Freq | Yield Freq |
> > > > |---------|---------|---------|---------|---------|---------|---------|
> > > > | WOMD | 1.021 | 21.202 | 22.825 | 0.041 | 0.043 | 0.019 |
> > > > | WOMD_interactive | 0.824 | 19.559 | 21.748 | 0.042 | 0.049 | 0.024 |
> > > > | Argoverse | 0.813 | 20.161 | 22.190 | 0.034 | 0.022 | 0.033 |
> > > > | NuScenes | 1.032 | 25.046 | 24.172 | 0.029 | 0.013 | 0.021 |
> > > > | INTERACTION | 0.898 | 21.132 | 22.611 | 0.048 | 0.047 | 0.027 |
> > > > | L2T | **0.738** | **8.172** | **13.873** | **0.196** | **0.143** | **0.159** |

---

### Official Review · Reviewer_baVM · 2024-11-04

**Soundness:** 2
**Presentation:** 2
**Contribution:** 2
**Rating:** 5
**Confidence:** 5

**Summary:**

This paper leverages large language models (LLMs) to generate trajectories for autonomous driving by introducing an interaction-behavior-trajectory translation approach, rather than the commonly used interaction-trajectory translation in other LLM-based methods. Their approach consists of two stages: in the first, vehicle interaction descriptions are translated into behaviors, and in the second, these behaviors are converted into specific motion parameters to create trajectories. Additionally, the authors developed a new Language-to-Trajectory (L2T) dataset. Their method demonstrates promising performance on both the L2T and Waymo Motion Prediction datasets, outperforming state-of-the-art methods.

**Strengths:**

1. They have published a dataset , Language-to-Trajectory (L2T), which includes 240K textual descriptions of vehicle interactions and behaviours.

**Weaknesses:**

The clarity of the paper can be improved. In the introduction, the motivation and objectives should be clearly stated, along with a summary of their accomplishments.

It is unclear how the trajectories are generated. Are they based on real data, generated through a simulator, or did the authors develop their own simulator?

The methodology section is difficult to follow and needs rewriting for better clarity.

The paper does not specify what kind of input was used to train the LLM model. Did they use generated prompts, or did they provide feature embeddings?

**Questions:**

The clarity of the paper can be improved. In the introduction, the motivation and objectives should be clearly stated, along with a summary of their accomplishments.

It is unclear how the trajectories are generated. Are they based on real data, generated through a simulator, or did the authors develop their own simulator?

The methodology section is difficult to follow and needs rewriting for better clarity.

The paper does not specify what kind of input was used to train the LLM model. Did they use generated prompts, or did they provide feature embeddings?

---

> ### Author Response · Authors · 2024-11-20
> **(1/2) Thanks for your valuable comments!**
>
> We sincerely thank you for your valuable comments. Below, we respond to your questions point-to-point.
>
> **Q1:** The clarity of the paper can be improved. In the introduction, the motivation and objectives should be clearly stated, along with a summary of their accomplishments.
>
> **Answer:** We sincerely thank you for your valuable suggestion. In the revised introduction section (see the third paragraph),  we have compared the language interface based on the large language model (LLM) to the graphical and programmable interfaces for generating vehicle trajectories. The language interface with LLM can leverage the short text of vehicle interactions to generate vehicle trajectories with detailed motion parameters. It means that users can rapidly achieve a large amount of trajectory data, thus saving data collection efforts. This efficiency motivates us to study LLM as the trajectory generator.
>
> The objective can be found in the last sentence of the fourth paragraph of the revised introduction. In this paragraph, we have discussed the shortcomings of the existing language interfaces. Without the guidance of human beings' driving logic, these  language interfaces conduct "interaction-trajectory" translation, which means the text description of vehicle interactions is directly translated into trajectories. They likely generate vehicle trajectories with unreasonable driving behaviors. Moreover, they likely fail to generate diverse trajectories according to the unseen maps, lacking the data diversity centered on the trajectory generation task. Thus, the primary objective of this paper is to involve the human driving logic to guide the translation between the text description of vehicle interactions and trajectories.
>
> In the last paragraph of the revised introduction, we summarize our contributions as:
> - We advocate a new paradigm of the language interface with the“interaction-behavior-trajectory.
> - We propose a novel language interface, Traj-LLM. We also contribute a new L2T dataset containing 240K traffic scenarios with vehicles' interactive trajectories and rich text descriptions.
> - We use the vehicle trajectories generated by Traj-LLM, which is trained on the L2T dataset, to finetune the trajectory prediction models, whose performances are effectively improved on the public Waymo and Argoverse datasets. These results can further inspire relevant research on the language-based trajectory generator.
>
> **Q2:** It is unclear how the trajectories are generated. Are they based on real data, generated through a simulator, or did the authors develop their own simulator?
>
> **Answer:** Our simulator integrates the re-developed hardware and software to efficiently collect and annotate the vehicle trajectories in the L2T dataset. We have added more details about our simulator to Section A.1 of the revised supplementary file.
>
> The hardware component of our simulator mainly includes a steering wheel, a gear shifter, gas and brake pedals, and several displays, as shown in Figure 7(a) of the supplementary file. The displays visualize the first-person view of the road environments from the left, right, front, and back car sides, as shown in Figure 7(b). Here, the road environments are simulated by LGSVL, a simulation engine for autonomous driving. Our hardware simulates a realistic car cockpit, helping users to drive on the virtual road to produce vehicle trajectories as real as possible.
>
> The software component is mainly built on top of the LGSVL engine. LGSVL can record all trajectories of vehicles traveling on the virtual road. We re-develop this engine to enable the connection among multiple sets of the above hardware, thus allowing multiple users to control different cars on the same virtual road. This software facilitates the simulation of vehicle interactions like those in the real world. In Figure 8, we show the interactive cars in the virtual road from the third-person view.
>
> In Section A.3 of the revised supplementary file, we have compared the data distribution of our L2T dataset with other real-world datasets like Waymo. In Figure 12, we compare the distributions of trajectories' shapes represented by the waypoints' coordinates. The distributions are similar, showing that the vehicle trajectories in the L2T dataset are realistic.
>
> **Q3:** The methodology section is difficult to follow and needs rewriting for better clarity.
>
> **Answer:** We sincerely appreciate your feedback on our presentation. We respectfully invite you to specify which part of the methodology section is difficult to understand. This will help us to reorganize the content and make it more understandable. Additionally, we have provided an anonymous GitHub repository (https://github.com/anonymous-github-Traj-LLM/Traj-LLM) for the complete implementation of Traj-LLM to help readers to follow our work.

---

> ### Author Response · Authors · 2024-11-20
> **(2/2) Thanks for your valuable comments!**
>
> **Q4:** The paper does not specify what kind of input was used to train the LLM model. Did they use generated prompts, or did they provide feature embeddings?
>
> **Answer:** Llama-7B is the fundamental LLM in our implementation (see Section C of the supplementary file). A training sample for LLM can be regarded as a triplet (interactions, behaviors, trajectories). In this triplet, vehicle interactions and behaviors are text descriptions (see examples in Figure 4(c) and (d)). During the training, LLM takes input as the short text of  interactions, learning to parse the interactions and outputting more detailed text of behaviors. The output behaviors and the ground truth provided in the training sample are used to calculate the loss. Next, LLM takes input as the behaviors to output the trajectories with numerical coordinates, which are used to compute the loss with the ground truth in the triplet sample.

---

> ### Author Response · Authors · 2024-11-26
> **Please read our responeses and provide specific comments.**
>
> Again, we thank Reviewer baVM for the review. In the last round of review and rebuttal, we have further explained the input data and the overall workflow of our approach (see our answer to Reviewer baVM's **Q4**). We have also respectfully invited Reviewer baVM to specify which part of the methodology section is difficult to understand (see our answer to Reviewer baVM's **Q3**). Given that Reviewer baVM's question **Q3** about the missing details of the proposed approach is unclear, and now Reviewer baVM has repeated this question again, we kindly ask for Reviewer baVM's specific comment on  **which part of the methodology section is difficult to understand**. This definitely will be critical for improving our paper. Here, we express our greatest gratitude to Reviewer baVM's help!
>
> We are pleased that Reviewer baVM found the additional information in the supplementary material useful. However, it is infeasible to include all the details in the main paper, which has a 10-page limit. Again, we want to emphasize that we are always willing to modify and provide details as long as the reviewer can give us specific comments. If required, we can exchange the necessary contents of the supplementary file and less important contents of the main paper.

---

> ### Author Response · Authors · 2024-12-03
> **Thanks for your reviewing!**
>
> Dear Reviewer baVM,
>
> Again, we want to express our sincere thanks for the insightful feedback you have provided. Though the discussion deadline is approaching, we hope to receive your valuable comments on our paper presentation. Undoubtedly, your suggestion is critical for the success of this research work.
>
> Thanks again for your time and effort in reviewing our paper.
>
> Best,
>
> Authors of Paper ID 4151

---

### Meta-Review · Area_Chair_iCnr · 2024-12-23

**Metareview:**

The present work proposes a LLM-based approach that takes textual descriptions of driving scenarios as input and generates the corresponding vehicle trajectories. Additionally, the work introduces a dataset of textual descriptions with corresponding vehicle trajectories. The authors show that using such generated data can improve the performance of downstream prediction models on public trajectory prediction benchmarks. The main strength mentioned by the reviewers is the introduction of a new dataset, and the work is easy to follow. The remaining concern is that several clarifications have been missing or are located in the supplementary materials. Ultimately, most reviewers see the work as marginally on the accept side. Given the value of the dataset and the proposed language-based generation idea for the community, I believe it to be useful, but I would ask the authors to slightly update the manuscript in accordance with the comments below. The paper should also outline in detail the dataset generation process (ideally providing the source code for dataset generation as well).

**Additional Comments On Reviewer Discussion:**

The authors have extensively responded to the reviewers concern and raised two concerns with the AC. First, a concern about the review of reviewer 9Ava about the diversity. The reviewer still rates the paper on the accept side but is otherwise not required to be satisfied with the responses to another reviewers comments even if this comment has led them to lower their score and even if the other reviewer is satisfied with the response. As for the questions to reviewer baVM, this reviewer has provided the following list of unexplained points in the AC/reviewer discussion, which I would like the authors to consider as they prepare the final:

1. Figure 2 shows interactions, but it is unclear how these interactions were obtained. Were they labeled manually? If so, which protocol was used? If not, what method was employed to identify these interactions? The methodology section does not provide any explanation for this process.
2. Figure 2 illustrates the map as an input entity, but the format of the map is not specified. Was it represented as a bird’s-eye view (BEV), a point cloud, a 2D image, or an occupancy map? The text only vaguely refers to a “map of polylines.” Additionally, the description “M is the number of polylines in the map M. The initial trajectories and the map together represent the spatial configuration of vehicles on the map” is ambiguous. Are M and the map the same, or are they distinct entities? A clear explanation is necessary.
3. The term "random locality attention" is introduced but not defined, leaving readers without a clear understanding of its role or significance.
4. Figure 2 in the methodology section depicts an LLM model, but the text does not specify which LLM model was used. There is no discussion of how this model was chosen or why it is suitable for the task. 5.Section 4 introduces the L2T dataset but does not provide details about its origin. Was this dataset derived from an existing benchmark, or was it collected by the authors? The lack of clarity regarding how this dataset was created or sourced undermines its reliability. Many critical details mentioned above are missing from the main paper. These elements are essential for understanding the work and should be included in the primary content rather than relegated to supplementary materials

---

### Decision · Program_Chairs · 2025-01-22

Accept (Poster)